# A Synthesis of Three Decades of Hydrological Research at Scotty Creek, NWT, Canada

William Quinton[1], Aaron Berg[2], Michael Braverman[1,3], Olivia Carpino[1], Laura Chasmer[4], Ryan Connon[1], James Craig[5], Élise Devoie[1,5], Masaki Hayashi[6], Kristine Haynes[1], David Olefeldt[7], Alain Pietroniro[8], Fereidoun Rezanezhad[9], Robert Schincariol[10], Oliver Sonnentag[11]

[1]Cold Regions Research Centre, Wilfrid Laurier University, Waterloo, Ontario, Canada
[2]Department of Geography, University of Guelph, Guelph, Ontario, Canada
[3]GHD Canada, Waterloo, Ontario, Canada
[4]Department of Geography, University of Lethbridge, Lethbridge, Alberta, Canada
[5]Department of Civil and Environmental Engineering, University of Waterloo, Waterloo, Ontario, Canada
[6]Department of Geoscience, University of Calgary, Calgary, Alberta, Canada
[7]Department of Renewable Resources, University of Alberta, Edmonton, Alberta, Canada
[8]National Hydrology Research Centre, Saskatoon, Saskatchewan, Canada
[9]Water Institute and Department of Earth and Environmental Sciences, University of Waterloo, Waterloo, Ontario, Canada
[10]Department of Earth Sciences, Western University, London, Ontario, Canada
[11]Département de Géographie & Centre d'études nordiques, Université de Montréal, Montréal, Québec, Canada

*Correspondence to*: W. Quinton (wquinton@wlu.ca)

**Abstract.** Scotty Creek, Northwest Territories (NWT), Canada, has been the focus of hydrological research for nearly three decades. Over this period, field and modelling studies have generated new insights into the thermal and physical mechanisms governing the flux and storage of water in the wetland-dominated regions of discontinuous permafrost that characterises much of the Canadian and circum-polar subarctic. Research at Scotty Creek has coincided with a period of unprecedented climate warming, permafrost thaw, and resulting land cover transformations including the expansion of wetland areas and loss of forests. This paper 1) synthesises field and modelling studies at Scotty Creek, 2) highlights the key insights of these studies on the major water flux and storage processes operating within and between the major land cover types, and 3) provides insights into the rate and pattern of the permafrost thaw-induced land cover change, and how such changes will affect the hydrology and water resources of the study region.

# 1 INTRODUCTION

The circum-polar region of the northern hemisphere is warming more rapidly than any other on Earth (Richter-Menge et al., 2017). This diverse region contains extensive coverage of boreal forest, taiga, tundra, polar desert, glaciers, wetland, and open water including some of the largest lakes, rivers and deltas in the world, all of which are changing in various ways and degrees in response to climate warming. Many of these circum-polar cover types are the focus of current research on how their form and functioning is changing in response to unprecedented climate warming. Such research is greatly advanced where long-term (i.e. multi-decadal) study sites are present and their associated data archives document the nature and pattern of change, and can be used to quantify rates of change and predict change trajectories. Within the circumpolar region, northwestern Canada stands out as particularly sensitive to climate warming given the scale and rapidity of recently observed warming-induced biophysical changes. The lower Liard River valley of northwestern Canada has a sub-arctic climate, lies within the continental high boreal and discontinuous permafrost zones (Hegginbottom and Radburn, 1992), and is characterised by approximately 53,000 km$^2$ of flat, organic terrain with a high density of open water and peatlands (Aylsworth and Kettles, 2000). The lower Liard River valley is one of several extensive areas of permafrost-affected peatlands within the circum-polar region. Others include the extensive wetland areas of the Hudson Bay Lowlands, northern Fennoscandia, and western Siberia. The intensive and coordinated field and modelling studies that began at Scotty Creek in the lower Liard River valley in the mid-1990s and continues to the present day, coincided with a period of rapid climate warming and warming-induced environmental changes. The accumulated body of research at Scotty Creek presents a unique opportunity to evaluate the impacts of unprecedented climate warming on a widely-occurring circum-polar landcover type and on the distribution and routing of water within it. This paper therefore synthesises three decades of hydrological research at Scotty Creek to provide a comprehensive understanding of the hydrological functioning of permafrost-affected peatland terrain and how it is changing with on-going climate warming.

Scotty Creek (61°180N, 121°180W), 50 km south of Fort Simpson, Northwest territories (NWT), Canada has a continental climate with short, dry summers and long, cold winters (MSC, 2017). Over recent decades its climate has warmed at a rate twice the global average (Cohen et al., 2014; Richter-Menge et al., 2017) and is one of the most rapidly warming on Earth (Vincent et al., 2015). There is growing evidence in the scientific literature (e.g. Hinzman et al., 2013; St. Jacques and Sauchyn, 2009; Walvoord and Kurylyk, 2016; Walvoord and Striegl, 2007) that this warming is altering hydrological flux and storage processes with potential long-term consequences for the region's water resources. The mean annual air temperature (MAAT) for the period 1981–2010 at Fort Simpson is -2.8°C, with a mean January temperature of -24.2°C and a mean July temperature of 17.4°C. The MAAT has been rising rapidly in the region (2.5°C from 1950 to 2015) with the most pronounced increase in winter (4.5°C from 1950 to 2015) (Vincent et al., 2015). Mean annual precipitation (1981–2010) is 390 mm, with 149 mm (38%) falling in the form of snow. Annual total precipitation has remained relatively stable

over the past 50 years. Snowmelt usually commences in the second half of March and continues throughout most of April so that by May, only small amounts of snow remain (Hamlin et al., 1998).

Scotty Creek drains a nearly continuous cover of peat overlying a thick clay to silt-clay deposit of low permeability (Aylesworth and Kettles, 2000). The peat cover helps to preserve permafrost owing to the large thermal offset created by a dry, insulating peat surface soil layer (Camill and Clark, 1998). The occurrence of discontinuous permafrost in the lower Liard River valley and elsewhere in circum-polar region at this latitude, is therefore largely confined to the areas of high peatland coverage, with the remaining land-cover being largely permafrost-free (Figure 1a). Scotty Creek, in addition to its neighbouring basins of Manners Creek, the Birch, Blackstone and Jean-Marie rivers, all drain peatland-dominated land covers with discontinuous permafrost that typify not just the lower Liard River valley, but also the southern Taiga Plains and Boreal Plains ecoregions of Canada, and much of the circumpolar subarctic. In the 1990s, these basins were the focus of research in support of the Mackenzie GEWEX (Global Energy and Water Exchange) Study (MAGS) which aimed to improve the understanding of and ability to predict the flux and storage of water within and from the major biophysical land cover types of the Mackenzie River basin (Woo, 2008). MAGS researchers in collaboration with government agencies expanded precipitation and stream gauging networks in the lower Liard River valley, where they also improved the understanding of the water sources and pathways giving rise to stream hydrographs (Gibson et al., 1993). This led to advancements in the ability to simulate the hydrological response of basins in this region (e.g. Pietroniro et al., 1996; Hamlin et al., 1998). These studies were focused mainly on isotope geochemistry, satellite remote sensing and the evaluation of the performance of an existing hydrological model (WATFLOOD) for simulation of basin runoff. As such, they involved relatively little field-based investigation of hydrological processes.

In 1999, the emphasis of hydrological research in the lower Liard River valley shifted toward field-intensive studies, and the headwater area of the 152 km$^2$ Scotty Creek basin emerged as the focus of this work. In that year, the first multi-year instrumentation was installed at Scotty Creek for long-term monitoring of ground temperatures, soil moistures, water levels and precipitation. The Scotty Creek headwater area (hereafter "Scotty Creek") refers to the upper half of the basin (Figure 1b) that is underlain by discontinuous permafrost (Hegginbottom and Radburn, 1992) and blanketed with a continuous cover of peat. These characteristics are typical of the 'continental high boreal' wetland region (NWWG, 1988), where peat accumulations in the range of 2 m (McClymont et al., 2013) to 8 m (Braverman and Quinton, 2016) typically overlie a clay/silt-clay glacial till deposit of low permeability (Aylsworth and Kettles, 2000).

After more than 20 years, this paper recounts the history of hydrological and related permafrost field studies at the Scotty Creek Research Station (see: www.scottycreek.com), from its inception to the present time. It is shown how field-based studies over this period formed the basis of new conceptual models, which in turn shaped the development of numerical simulations of the hydrological processes for this environment. This paper also explains how the field and modelling studies

at Scotty Creek have contributed to research programmes elsewhere, and generally to cold regions hydrology and related disciplines. The following discussion begins with a characterisation of the landscape and description of the hydrological functioning of its major components. The properties and hydrological functioning of the active (i.e. seasonally-frozen) layer are then discussed, which provides the background for the subsequent discussion on runoff generation. The following two sections examine the hydrological impact of permafrost thaw, first in relation to climate warming induced land cover change, and then in relation to linear disturbances, the most widely occurring form of direct human disturbance in the study region. This is followed by a discussion of the impact of Scotty Creek research on the wider scientific community. The final section discusses on-going research and future studies at Scotty Creek.

## 2 LANDSCAPE CHARACTERISATION AND FUNCTIONING

At the time field studies began at Scotty Creek, the hydrological function of such low relief, peatland-dominated zones of discontinuous permafrost was poorly understood. This lack of understanding was problematic, in large part, because it prevented the development of physically-based hydrological models for reliable prediction of runoff from a terrain type that covers approximately 23% of all permafrost soils in the discontinuous/sporadic zones in North America, approximately 19% of all permafrost soils in the circum-polar north (Tarnocai et al., 2009) and dominates much of the Canadian subarctic. It was also problematic because this terrain is a critical component to the global carbon balance (Grosse et al., 2011; Schaefer et al., 2011), and without a sufficient understanding of its hydrological functioning, its role in climate warming is uncertain.

Initial ground-based and remote sensing studies (Quinton et al., 2003) indicated that Scotty Creek is dominated by forested peat plateaus that are underlain by permafrost and rise 1 to 2 m above permafrost-free, treeless wetlands in the form of collapsed wetlands and channel fens. Cross-sections of permafrost bodies at Scotty Creek measured from electrical resistivity imaging indicate they are on the order of 10 m thick with near vertical sides (McClymont et al., 2013). The plateaus and collapsed wetlands are arranged into distinct complexes separated by channel fens. The National Wetlands Working Group (1988) refers to the collapsed wetlands as "collapse scars" and describes them as resulting from thermokarst erosion, a process that transforms peat plateaus to flat, low-lying, treeless and permafrost-free wetlands. In earlier publications on Scotty Creek, the collapsed wetlands were referred to as "flat bogs" to distinguish them from the more common domed bogs of lower latitudes. However this term became problematic since by definition, bogs receive hydrological input from precipitation only, and subsequent research at Scotty clearly demonstrated that the "flat bogs" receive substantial lateral runoff from adjacent plateaus.

The contrasting biophysical characteristics among the three land cover types (peat plateaus, collapsed wetlands and channel fens) led to the proposition that each has a specific function in the basin water balance (Quinton et al., 2003; Hayashi et al., 2004) (Figure 2). Since peat plateaus rise above the surrounding wetlands, support a relatively deep snow cover, and have a

limited capacity to store hydrological inputs, these were seen to function mainly as runoff generators (Figure 3a). In contrast, because collapsed wetlands were thought to be entirely surrounded by the raised permafrost of the plateaus, water entering these wetlands was assumed to remain in storage until evaporated or displaced into the groundwater system. In this sense, the plateaus were seen as "*permafrost dams*". In a later paper, Quinton et al. (2009a) distinguished between collapsed

wetlands that occur within peat plateaus and are hydrologically-isolated from channel fens, and those that are situated between plateaus and have an open connection with channel fens. Although isolated collapsed wetlands are large in number, they account for less than 5% of the Scotty Creek catchment, while the area occupied by connected collapsed wetlands is more than five times larger and roughly equivalent to the area occupied by channel fens (Quinton et al., 2009a). This conceptualisation of the form and hydrological functioning of collapsed wetlands was modified again by Connon et al.

(2015) who demonstrated that many of these features that were assumed to be hydrologically isolated can form ephemeral connections with a fen (either directly or via another collapsed wetland or series of them) during periods of high moisture supply, such as during the snowmelt runoff period. (A more detailed discussion on the development and hydrological functioning of such connections is presented below in the context of permafrost thaw). Water draining from plateaus into channel fens is conveyed downstream toward the basin outlet, as the primary hydrological function of these features is lateral

flow conveyance along their broad, hydraulically-rough channels. The water level response to summer rainstorms measured at several nodes along the main drainage network at Scotty Creek showed that the flood-wave velocity is controlled by channel fen slope and hydraulic roughness in a way consistent with the Manning formula, suggesting that a roughness-based algorithm might be useful for routing water at the basin scale (Hayashi et al., 2004).

The contrast between the mainly storage function of the collapsed wetlands and the conveyance function of the channel fens, suggests that the relative proportion of these two cover types in a basin would influence the volume and timing of the outflow hydrograph. To investigate this possibility, the runoff response of Scotty Creek and four neighbouring basins (Jean-Marie, Birch, Blackstone and Martin Rivers) was examined by Quinton et al. (2003) for the four year period 1997 to 2000 in relation to the relative proportions of peat plateau, collapsed wetland and fen, and other biophysical characteristics. This

study lent support to the proposition that channel fens are primarily water conveyers and that collapsed wetlands predominantly store water since it showed that runoff was positively correlated with the percentage of the basin covered by channel fens, and negatively correlated with the percentage of the basin covered by collapsed wetlands. Although that study led to a new conceptual understanding of the hydrological functions of the major terrain types, the hydrological interactions among these terrains remained poorly understood. However, the study by Hayashi et al. (2004) shed light on such

interactions using isotope geochemical methods. Specifically, they showed that Goose Lake at the headwater of Scotty Creek had a strongly enriched isotopic composition, but that the stream water composition was gradually depleted over the 14 km flow distance from Goose Lake to the Scotty Creek outlet. This indicated a continuous hydrological connection between Goose Lake and the basin outlet, and that over this distance lateral drainage from peat plateaus contributed isotopically light and chemically dilute water to channel fens. This was supported by measurements of electrical conductance which increased

with increasing distance from the edge of plateaus, suggesting that the plateaus were a source of chemically-diluted water (Quinton, unpublished data, 2004). The authors also found that later in summer when water levels were relatively low, Goose Lake became hydrologically disconnected from the basin outlet.

Hayashi et al. (2004) considered collapsed wetlands to be hydrologically-isolated as described by Quinton et al. (2003), however they noted that such wetlands abutting fens without an intervening permafrost barrier could also contribute to a fen the water they received from precipitation or as runoff from their adjacent peat plateaus. Chemical and isotopic analysis of surface and subsurface waters in the Scotty Creek basin indicated that snowmelt water contributed less than half of the basin discharge during the spring freshet, indicating that a large amount of water must be stored overwinter, presumably in the

lakes, channel fens and their abutting collapsed wetlands (Hayashi et al., 2004). The total amount of water stored over the winter in the basin was estimated to be 140 to 240 mm, which was comparable to the average annual basin discharge (149 mm). The average evapotranspiration estimated from chloride mass balance computations was 280 to 300 mm per year, which was consistent with the value of 275 mm per year estimated from precipitation minus runoff. The results presented by Hayashi et al. (2004) are consistent with those of Gibson et al. (1993), who found that only 40 to 50% of peak runoff in

Manners Creek was supplied by snowmelt.

Collectively, these studies helped to define the hydrological functions of the land cover types that predominate throughout much of the peatland-dominated zone of discontinuous permafrost. These studies also demonstrate the unique hydrological behaviour of sub-arctic peatland complexes compared to their counterparts in warmer climates. Much of this unique

functioning is due to the juxtaposition of permafrost (plateau) and permafrost-free (wetland) land covers, and to the profound hydrological impact of saturated, and therefore relatively impermeable permafrost whose upper surface (i.e. permafrost table) rests above the elevation of the adjacent permafrost-free wetland ground surfaces. This elevational difference produces a hydraulic gradient driving flow from the plateaus toward the wetlands, and prevents wetlands (surrounded by raised, saturated permafrost) from draining. As such, there exists in this subarctic environment, a source-sink relationship between

plateaus and wetlands with respect to runoff. As noted however, exceptions occur where permafrost thaw lowers the permafrost table to an elevation below that of the wetland water table, in which case the wetland sheds water by surface and near surface flow. This process initiates preferential thaw of permafrost at the wetland outlet and below the drainage pathway leading from it. Preferential permafrost thaw of this type is typically coupled with the formation of suprapermafrost taliks which enable wetland drainage throughout the year. Plateaus also control drainage at the larger "landscape" scale

depicted in Figure 2 where water flowing through channel fens and other hydrologically-connected wetlands, must flow around plateaus. As permafrost thaw progresses and the cover of plateaus diminishes, the extent to which plateaus obstruct and re-direct flow over the landscape would also decrease. The contrasting hydrological functions of the plateau, collapsed wetland and fen are accentuated by the presence of permafrost. Indeed without permafrost, there would be no plateaus, and redistribution of water at the local scale by the sink-source relation referred to above, and the redirection of water at the

larger landscape scale would not occur. For example, in the permafrost-free Utikima region of northern Alberta, Canada, the hydraulic gradient between raised peatlands and the surrounding wetland terrain is opposite in direction to what is observed at Scotty Creek. The relatively low water table below the raised peatlands of Utikima resulting from the relatively high evapotranspiration rates of their tree covers causes water to flow toward the raised peatlands, rather than away from them as observed at Scotty Creek (Devito et al., 2017).

## 3 ACTIVE LAYER PROPERTIES AND FUNCTIONING

By characterising the hydrological functions of the major land cover types (Quinton et al., 2003) and interactions between them (Hayashi et al., 2004); a new conceptual framework for runoff generation in wetland-dominated terrain with discontinuous permafrost had emerged. Such a conceptual framework was an essential step toward the development of a hydrological model for this terrain type. However, field studies at Scotty Creek also focused on smaller scale phenomena driven by the need to improve the process understanding and parameterisation of the physical, hydraulic and thermal properties of the peat blanketing the collapsed wetlands, fens and plateaus. For example, Quinton and Gray (2001) showed that estimation of subsurface flow from organic-covered, permafrost terrain requires that the elevation and thickness of the saturated layer be known because the peat permeability can decrease by 2 to 3 orders of magnitude between the ground surface and 40 cm depth. Since the saturated layer is perched on the relatively impermeable frost table, its elevation decreases and thickness increases as the ground thaws and the frost table lowers. Therefore, estimating the rate of subsurface runoff requires not only proper characterisation of peat hydraulic properties, but also knowledge of the degree of thaw of the active layer (i.e. frost table depth). This early work by Quinton and Gray formed the conceptual basis of the numerical model developed and applied by Wright et al. (2009) to route subsurface drainage over the topographic surface of the frost table. This model is discussed in more detail below.

Quinton et al. (2000) reported that the values of total porosity decreased only slightly with depth, from approximately 95% in the 0-5 cm zone to 85% at 35 cm depth. However, the authors reported that over this depth range, values of 'active porosity' (i.e. the proportion of the total peat pore volume that actively transmits water (Romanov, 1968)) typically decreases from approximately 80% to <50%. The authors measured active porosity from image analysis of 2D thin sections of peat samples, whereby the inter-particle area expressed as a percentage of the total image area was assumed to approximate the active porosity. Quinton and Hayashi (2004) also demonstrated that the fraction of the total porosity that conducts water decreases with increasing depth, although these authors did so through drainage experiments. They reported that the 'drainable porosity' (i.e. the pore volume of water removed when the water table is lowered) decreases from approximately 0.6 near the ground surface to 0.05 at 40 cm depth. These relatively low values, especially at depth, enable a rapid response of the water table to hydraulic inputs to the ground surface. This rapid response is enhanced by very high infiltration rates and the close proximity of the saturated zone to the ground surface. Laboratory drainage tests combined with microscopic

image analyses indicate that with increasing depth, the proportion of small, closed and dead-end pores increases as does the water content for a given pressure, and that the peat maintains a residual volumetric moisture content of 15-20% (Quinton and Hayashi, 2004).

Flow through peat is typically evaluated using macro-scale concepts such as the hydraulic conductivity, with little regard for the microscale properties of pore size, geometry and connectivity. Characterizing the depth variations of these microscale properties in peat profiles makes an important contribution to the literature on peat hydrology because it increases the physical understanding of and ability to predict hydrological and biogeochemical fluxes through peat and peatlands, and how such fluxes might change in response to disturbances that change these properties. An early example of the research at Scotty
Creek that contributed new knowledge on the physical and hydraulic properties of peat at the pore scale is the study by Quinton and Gray (2001) which showed that the dimensionless coefficient $C$ of the relationship between friction factor, $f$ and Reynolds Number, $N_R$, (i.e. $f = C/N_R$) increased linearly with the depth to the middle of the saturated zone, $d$ [L]. Since $C = 2D^2/k$, where $D$ [L] is the geometric mean pore diameter of the material encountered by the saturated layer, and $k$ [L$^2$] is the intrinsic permeability, the relationship between $C$ and d allows an approximation of the variation in permeability with depth
(Quinton and Gray, 2001). In a later study, Quinton et al. (2008) focused on the saturated hydraulic conductivity, $K$ [LT$^{-1}$], the product of $k$ and $\rho g/\mu$, where $\rho$ [ML$^{-3}$] is the density of water, $g$ [LT$^{-2}$] is the gravitational acceleration, and $\mu$ [ML$^{-1}$T$^{-1}$] is the dynamic viscosity. Three independent measures of $K$ were used to explore how its value varies with depth: tracer tests, constant-head well permeameter tests, and laboratory measurements of undisturbed samples. The conductivity profiles contained very high values (10-1000 m d$^{-1}$) within the top *ca.* 0.1 m where the peat is only lightly decomposed, a large
reduction with increasing depth below the ground surface in the transition zone, and relatively low values in a narrow range (0.5-5 m d$^{-1}$) below *ca.* 0.2 m depth, where the peat is in an advanced state of decomposition (Figure 4). Digital image analysis of resin-impregnated peat samples showed that $K$ is essentially controlled by pore hydraulic radius, which decreases with depth due to increasing compaction by overlying sediments. The Quinton et al. (2008) study benefitted from the work of Hayashi and Quinton (2004), which extended the applicability of the Guelph permeameter method to the soil conditions
of Scotty Creek (i.e. relatively thin soil overlying impermeable permafrost) and produced a new set of shape factors determined by numerical simulation.

The research at Scotty Creek on the physical and hydraulic properties of peat discussed above was then augmented by a series of studies that made use of innovative analytical laboratory methods. As such, these studies made additional
contributions to the literature by demonstrating new pore-scale visualisation and measurement techniques. For example, high-resolution X-ray tomographic images were used to elucidate the volume and configuration of the pore network from both two and three dimensions for discrete ranges of soil water pressure typically observed at Scotty Creek. The active porosities measured in 2D and 3D were very similar, and the volumetric moisture content measured using the tomographic images closely approximated the gravimetric measurements (Quinton et al., 2009b). This volume loss was accommodated by

the thinning and disaggregation of moisture films, although as the sample lost moisture, the flow network maintained a relatively even spatial distribution. In related studies, Quinton et al. (2009b) and Rezanezhad et al. (2009; 2010) used X-ray Computed Tomography (CT) to visualize the pore structure of peat from a peatland-dominated zone of discontinuous permafrost at Scotty Creek. They found that the pore distribution in the near-surface sample of peat was dominated by a

single, large highly connected and complex pore space which accounts for 94-99% of the total inter-particle pore volume. Analysis of the tomographic images also revealed that with increasing depth, the pore size and the degree of interconnection among pores decreased rapidly while the number of pores and the tortuosity of the pathways connecting them increased. In subsequent studies using tomographic images (Rezanezhad et al. 2012; 2016), the authors showed the complex dual-porosity nature of peat with active (i.e. mobile) and inactive (i.e. immobile) regions within the pores, which can be an important

factor in water storage, flow, and solute migration. Gharedaghloo et al. (2018) re-analysed the CT scan datasets used by Quinton et al. (2009b) and Rezanezhad et al. (2009; 2010) to derive pore connectivity, pore radii variation, and pore tortuosity and used these values to compute $K$ of peat and its variations with depth. The authors explained the reduction of $K$ with depth as the cumulative result of decreasing pore radii and decreasing pore tortuosity, both driven by increasing compaction by overlying peat with increasing depth. Through hydrological modelling, this study also showed that the pore

peat network is not inherently anisotropic (Gharedaghloo et al. 2018), but anisotropy results from upscaling due to variations of $K$ between peat layers.

The knowledge gained from the above studies on the physical and hydraulic properties of the peat mantling the plateaus was instrumental to understanding the hydraulic response of the landscape. However, it was also recognised through other studies

at Scotty Creek that the flux of water is closely coupled to the flux of energy. For example, Quinton and Gray (2001) showed that since $K$ decreases with depth, the average $K$ of the saturated layer of peat decreases with time as the relatively impermeable frost table lowers through the active (i.e. seasonally-thawed) layer as it thaws. Quinton and Gray (2003) demonstrated a strong correlation between cumulative degree-day ground surface temperature and the fraction of the cumulative ground heat flux used to melt ice in the active layer, and suggested that the former could be used to estimate the

frost table depth. Hayashi et al. (2007) developed a simple but effective heat-conduction model to simulate the downward movement of the impermeable frost table during thawing. Simulations were compared with the ground heat flux measured simultaneously using calorimetric, gradient, and flux-plate methods. The majority (86%) of incoming ground heat flux was used to melt the ice in the active layer. Simulated depths to the frost table during the 2003 to 2005 thaw seasons matched closely with observed data for contrasting ground-cover types.

The depth to the frost table was found to vary widely over short distances (Quinton and Gray, 2001), and as a result, so too does the topography of the frost table. Wright et al. (2009) used this premise to demonstrate that topographic variations of the relatively impermeable frost table control the rate and direction of subsurface flow, since the flow rate is a function of the depth-dependant value of $K$ and the frost table slope angle, while the flow direction is governed by the direction of the

sloping frost table. As such, the frost table topography includes areas of pooled water in frost table depressions and areas of preferential flow in frost table channels, although such features are often transient as the thaw season progresses and the frost table thaws differentially. Variations in soil moisture were found to be the dominant factor controlling depth to the frost table whereby wetter areas were associated with deeper frost tables and wetter years had greater average end of summer thaw depths (Wright et al., 2009). A recent study at Scotty Creek on the hydrological impacts of wildfire (Ackley, 2018) demonstrates close connections among thermal, physical and hydraulic properties of the active layer. This work showed that relative to an adjacent non-burned control site, a low-severity fire in July of 2014 increased the bulk density and moisture retention for a given level of applied negative pressure, and decreased porosity and hydraulic conductivity. They also demonstrated that the average end-of-season thaw depth increased while its spatial variability decreased. By reducing spectral heterogeneities of the ground surface, the fire reduced the spatial variations of ground thaw and therefore the topographic variations of the underlying frost table. This change was found to reduce preferential flow and storage of water at the burned site relative to an adjacent undisturbed site. Similarly, Gibson et al. (2018) examined active layer dynamics in the years to decades following fire events. The authors concluded that active layer depth increased by approximately 50% in the first decade after the fire, with convergence to the depth observed in unburned sites on the order of 20 years post-fire.

Researchers have often adapted for northern conditions, runoff generation concepts developed for temperate or other regions such as variable source area, transmissivity feedback, and fill-and-spill (e.g. Tromp-van Meerveld and McDonnell, 2006). However, based on the close coupling of energy to the hydrological cycle demonstrated at Scotty Creek, a new "energy-based" framework was developed for delineating runoff contributing areas for organic-covered, permafrost terrains (Wright et al., 2009; Quinton and Carey, 2008). Spatial variations of aerodynamic energy and roughness height affect the end-of-winter spatial distribution of snow (Pomeroy et al., 2004), while spatial variations of radiant energy control the spatial distributions of snowmelt and ground thaw rates. The combined spatial pattern of aerodynamic (i.e. turbulent) and radiant energy control the topographic variations of the relatively impermeable frost table. The features of the frost table, such as local areas of preferential thaw corresponding to areas of preferential subsurface storage or flow, were found to persist year-to-year.

The field studies of thaw rates and patterns described above were complemented by laboratory investigations on the interaction of peat with the frost table. For example, in 2007, four 0.6 m diameter, 0.75 m deep peat cores with 0.25 m surface vegetation were sampled from permafrost plateaus at Scotty Creek and transported to Western University (London, Ontario, Canada) where they were installed in a climate controlled chamber and instrumented with energy and mass flow sensors. The temperature and moisture gradients in the chambers were set to values measured in the instrumented soil pits established in 2003 (Table 1). These laboratory experiments, along with coupled heat and water transport numerical modelling, evaluated the sensitivity of soil freezing / thawing to variations in soil temperature and moisture, snow-cover thickness, radiation regimes, and mitigation measures to reduce the impacts of disturbance. Nagare et al. (2012) provided

insight into the field observed movement of frost tables upward during winter. Soil water movement towards the freezing front was inferred from soil freezing curves, liquid water content time series and from the total water content of frozen core samples. A substantial amount of water, enough to raise the upper surface of frozen saturated soil within 0.15 m of the soil surface at the end of freezing period appeared to have moved upwards from the permafrost zone. Diffusion under moisture

gradients and effects of temperature on soil matric potential appear to drive such movement (Kurylyk and Watanabe, 2013).

The knowledge arising from studies on active layer processes at Scotty Creek was applied to the development of new permafrost thaw remediation strategies of interest to local communities, government agencies and industry. For example, climate chamber and numerical studies by Mohammed et al. (2017) quantified the effects of mulching and its ability to limit

permafrost thaw and alterations to the ground thermal regime. Overall, the thermal buffering ability of the mulch had beneficial effects on slowing thaw due to its low thermal conductivity, which decouples the subsurface from meteorological forcing and impedes heat conduction. Aside from the dry ground-0.1 m mulch scenario which did not have a positive effect on reducing ground thaw, thaw reduction ranged from 12 to 75%, with the wet ground-0.3 m mulch scenario achieving the maximum thaw depth reduction. Results also suggested that mulching over aging disturbances, such as seismic lines where

permafrost is very degraded, may have the potential to stabilize thaw or even regenerate permafrost (Mohammed et al., 2017). Knowledge arising from these studies has been applied by the Government of the Northwest Territories to the development of best practices regulations designed to minimize the impact of disturbance to ground surfaces overlying permafrost. Permafrost thaw remediation research at Scotty Creek has also made progress in the development of new ground freezing systems for the purpose of slowing, arresting or even reversing permafrost (Braverman and Quinton, 2017). Initial

investigations used passive thermosyphons installed in a seismic line to a depth of 2 m (*i.e.* top of permafrost table). The cooling effect of passive thermosyphons is driven by the circulation within sealed thermosiphon pipes in response to a temperature gradient between the top (colder) and bottom (warmer) of the pipes, and by the consumption (bottom of pipe) and release (top of pipe) of latent heat. However, it was found that these thermosyphons had little effect on the subsurface temperatures which remained at the freezing point depression of -0.2$^{o}$ C. After several iterations of design, it was found that

single phase cooling pipes that circulate liquid coolant using specially designed submersible pump lowered subsurface temperatures to below -6$^{o}$ C (Braverman and Quinton, 2017). These remediation studies gained much interest from local communities, industry and government agencies interested in protecting infrastructure from damage due to permafrost thaw.

Collectively, the studies described in this section at Scotty Creek have advanced our understanding of the critical and unique

hydrological phenomena of the active layer. Much of this uniqueness is ascribed to the presence of discontinuous permafrost, seasonally frozen soil, and the distinctive thermal, physical and hydraulic properties of peat. The key contributions, addressed concurrently through field, laboratory, and modelling investigations, were the identification of energy as a driver of runoff, the recognition of the considerable role of the vertical variability of peat on hydrological response, and the role of the dynamic frost table topography on storing and shedding water from peat plateaus. These phenomena have since been

identified as important in other landscapes, including Prairie (Hayashi et al., 1998; Hayashi et al., 2003; Fang et al., 2010), subalpine (Carey and Woo, 2001; Carey et al., 2012; Pomeroy et al., 2003), taiga shield (Guan et al., 2010; Spence et al., 2010), arctic tundra (Liljedahl et al., 2016), and other subarctic regions of Canada (Devito et al., 2017; Ferone and Devito, 2004), Alaska (Yoshikawa and Hinzman, 2003) and Siberia (Brutsaert and Hiyama, 2012). As such, the application of

research discussed in this section extends well beyond Scotty Creek, not only to other wetland-dominated regions of discontinuous permafrost throughout the circum-polar region, but to other climatic regions and biophysical site types.

## 4 RUNOFF FROM PEAT PLATEAUS

Given the emergence of the conceptual framework for runoff generation described above, researchers at Scotty Creek focused their initial efforts on peat plateaus since these features were considered to be runoff generators. Wright et al. (2008)

used a water balance approach and the Dupuit-Forchheimer equation to quantify sub-surface runoff from a plateau at Scotty Creek and showed that 1) these two computations yielded similar results in both years (2004, 2005) of study, and 2) runoff accounted for approximately half of the moisture loss from the peat plateau, most of which occurred in response to snowmelt inputs. The melt of ground ice was also a significant source of water during the study periods, which was largely retained in soil storage. Soil moisture conditions prior to soil freezing were a major factor controlling the volume of runoff from the

hillslope. Subsurface drainage rates declined dramatically after the snowmelt runoff period, when the majority of water inputs were stored or lost to evapotranspiration. The minimal lag between rain events and subsurface runoff response in both years suggests that much of the runoff produced from rain events is rapidly transported to the adjacent wetlands.

Early studies at Scotty Creek reported that the frost table is typically at or near the ground surface by the end of winter,

suggesting that at the onset of ground thaw, the active layer is saturated or nearly saturated with ice and a small amount of unfrozen water (Quinton and Gray, 2001). Data from instrumented soil pits indicated that the unfrozen water content was ~15-20% (volumetric), roughly consistent with the residual moisture value reported by Quinton and Hayashi (2004). Considering that at the time of soil freezing in the autumn, the water table is often at a depth of 40 cm or more, a frost table position near the ground surface implies considerable over-winter moisture movement within the active layer. Analysis of

the soil cores removed from the ground in April, 2003 indicated that between freeze-up and late winter, the total (frozen and unfrozen) soil moisture increased throughout the active layer, and that this increase was greatest close to the ground surface. It was also found that the total soil moisture below 0.3 m depth was close to saturation. However, as the cumulative number of active layer thaw measurements at Scotty Creek increased throughout the decade, the assumption that the active layer is nearly or fully saturated by the end of winter was called into question. For example, Quinton and Baltzer (2013)

demonstrated that the observed rate of thaw was often greater than could be explained using numerical models and suggested incomplete freezing of the active layer during winter or the existence of a perennially-thawed layer (i.e. talik) that provides an additional source of heat for active layer thaw. Since that study was published, the widespread occurrence of taliks has

been demonstrated below linear (i.e. seismic) disturbances (Braverman and Quinton, 2016), in peatland areas 10-20 years following the occurrence of wildfire (Gibson et al., 2018) as well as below large areas of peat plateaus (Connon et al., 2018a).

The new knowledge and numerical descriptions of active layer thermal (Hayashi et al., 2007; Kane et al., 2001), physical (Quinton et al., 2008; Hayashi and Quinton, 2004) and hydraulic (Wright et al., 2008) properties and processes arising from the field studies at Scotty Creek was not an end in itself, but was applied to inform numerical model development so that the rates and patterns of ground thaw and hydrological fluxes could be more confidently predicted. Much of this new knowledge was incorporated into the Cold Regions Hydrological Model (CRHM). CRHM was then applied to address specific

questions as to a variety of spatial and temporal scales. For example, Quinton and Baltzer (2013) then applied CRHM to evaluate the impact of permafrost thaw induced plateau shrinkage and subsidence on plateau runoff production. At a larger scale, Stone *et al.* (In press) applied CRHM to examine the impact of permafrost loss in a 0.45 km$^2$ sub-catchment on runoff from an adjacent channel fen. Applications of models informed by knowledge arising from rigorous field studies are important diagnostics for evaluating the hydrological impacts of different scenarios of permafrost thaw-induced land-cover

change. The new knowledge on peat thermo-physical and hydraulic properties also formed the basis of a quasi-3D, coupled heat and water transfer model presented by Wright et al. (2009) to simulate active layer thawing and runoff generation in a plateau. This new model known as *SFASH* (Simple Fill and Spill Hydrology), applied a unique variation of the fill-and-spill runoff paradigm to plateau runoff, whereby water stored in topographic depressions of the frost table is released (i.e. 'spilled') once a storage threshold is exceeded by precipitation forcing, or by thaw-induced changes to the depression storage

capacity. Differential ground thaw makes this variation on the fill-and-spill runoff paradigm far more dynamic.

While Wright et al. (2009) considered relatively small (25 m$^2$) study plots along the sloping edges of peat plateaus (a part of plateaus designated as a *"primary runoff"* producing area in a later study by Connon et al., (2014) described below), Christensen (2014) simulated runoff processes over the entire area of peat plateaus. In doing so, the author modified the

Northern Ecosystem Soil Temperature (NEST) model of Zhang et al. (2003) and used it to provide the surface boundary conditions for two- and three-dimensional subsurface heat and water transfer models to simulate the vertical and lateral thawing of permafrost (McClymont et al., 2013; Kurylyk et al., 2016). A relationship was developed between plateau geometries and runoff timing. Using a dimensionless form of the Boussinesq equation, similar to *SFASH* model, it was shown that the runoff timing from plateaus is dependent on the height and depth of a plateau. Using this relationship, along

with the hydraulic radius to approximate plateau radius, the runoff timing from irregularly shaped plateaus can be calculated. Future efforts developing a better averaging technique for hydraulic conductivity and a more appropriate equivalent radius approximation are required for application of these methods over an entire basin. By combining the equations for runoff timing from individual plateaus developed in this study with a routing algorithm for moving water throughout the basin, the hydrological response of an aggregate of peat plateaus in the discontinuous permafrost zone could be determined.

In summary, the preceding studies found runoff from peat plateaus to be mostly in the shallow subsurface and controlled by antecedent moisture storage and melt of ground ice. The presence of local taliks influences this initial state, but in general, local runoff processes are now readily modelled via a number of techniques; challenges remain in determining what happens to this water once it runs off to adjacent fens and collapsed wetlands, and how these local flux and storage processes scale to a basin the size of the Liard River.

## 5 SCOTTY CREEK IN TRANSITION

The land cover at Scotty Creek is among the most rapidly changing on the planet (Camill, 2005; Osterkamp and Romanovsky, 1999). Over successive years, changes to the landscape, such as the expansion of wetlands and flooding of forests, were clearly evident and indicative of permafrost thaw. Given that black spruce (*Picea mariana*) forests at Scotty Creek exist only on areas underlain by permafrost (i.e. peat plateaus), the extent of permafrost and the rate of permafrost loss can be estimated from analysis of aerial or satellite images using forest cover as a proxy (Carpino, 2017). An investigation into the rate and pattern of permafrost loss from this method (Quinton et al., 2011) showed that the proportion of a 1 km$^2$ area underlain by permafrost decreased from 55% to 43% between 1970 and 2008. Although aerial photographs older than 1970 are available (e.g. 1947), their quality is low, and as a result so too is the confidence in drawing conclusions from their analysis. However, climate data recorded at Fort Simpson indicate that rapid warming in the region did not commence until the early 1970s (see Fig. 8a in Quinton et al., 2009a), and numerical simulations with the NEST model (Christensen, 2014) suggested relatively stable permafrost until approximately 1980. A subsequent study by Chasmer et al. (2011) presented new remote sensing methods of estimating permafrost distribution, including the use of light detection and ranging (LiDAR) techniques to precisely define permafrost plateau edges based on ground surface elevation. Other related studies demonstrated that thaw rates are accelerating due to fragmentation of the permafrost bodies (Veness, 2014; expanded upon by Baltzer et al., 2014). In the discontinuous permafrost zone, permafrost thaw involves simultaneous lateral recession of the near vertical sides (McClymont et al., 2013) of a permafrost body, and lowering of its permafrost table.

Connon et al. (2018) demonstrated that the development of talik is a kind of "*tipping point*" that greatly accelerates the rate of permafrost thaw. Specifically, they found that the permafrost below areas with a talik thawed five times faster than areas without a talik, as the unfrozen talik prevents energy loss from the permafrost body to the atmosphere during the winter. Similarly, Gibson et al. (2018) concluded that following wildfire, rates of lateral expansion of collapsed wetlands were particularly high where talik was already present below adjacent peat plateaus. Connon et al. (2018) also identified the critical thaw depths associated with talik development. For example, in early April 2016, the depth of re-freeze was measured at 135 points along nine transects that have been monitored for thaw depth and soil moisture since 2011. It was found that the average re-freeze depth was 65 cm. The minimum and maximum re-freeze depths were 60 cm and 80 cm

respectively (mean = 65 cm). The same results were found in replicate measurements in April, 2017. These findings indicate that if the ground thaws to a depth of 60 cm, it may not refreeze in the following winter, and if summer thaw reaches or exceeds 80 cm, it can be assumed that it will not entirely refreeze (Figure 5). These findings are supported by the long-term active layer monitoring at Scotty Creek. For example, Quinton et al. (2011) reported that the average end-of-summer thaw depth for a peat plateau at Scotty Creek for the period 1999 to 2004 was 0.58 m with little variation from year to year (SD = 0.04). However, following, 2004, the thaw depth increased by approximately 0.07 m yr$^{-1}$ such that by 2018 the average end-of-summer thaw depth had increased to over 1.6 m (Figure 6). The increase in vertical thaw after 2004 coincided with an increase in the rate of lateral shrinkage of the plateau on which the measurements were made. The introduction of a talik changes the suprapermafrost layer from a single layered (active layer) to a dual layered (active layer and talik) system. The establishment of a talik accelerates permafrost thaw because it reduces the loss of heat from the underlying permafrost during winter while introducing a second thawing front to the overlying active layer.

Just as permafrost thaw is changing the hydrological function of collapsed wetlands by introducing new drainage channels that connect them, it is also changing the hydrological function of the plateaus by introducing a talik, a new flowpath that conveys water throughout the year. Although taliks may be hydrologically-isolated in depressions of the permafrost table, others taliks extend across plateaus, hydrologically-connecting the wetlands on either side of them (Figure 7). In the case of the latter, if the permafrost table is lowered to below the elevation of the water table of the adjacent wetlands, then the plateau is no longer able to function effectively as a permafrost dam as water can be conducted over the permafrost table through the talik throughout the year. As such, taliks can introduce a new subsurface flow path, which in addition to the new surface pathways (e.g. see "*wetland capture*" below), contributes to the increase in basin hydrological connectivity. This process may have been responsible for the observed greater summer runoff from Notawhoka Creek (50 km east of Scotty Creek) than from Scotty Creek, due to the increased ground thaw depths within the Notawhoka Creek catchment that developed following an extensive wildfire in 2014 (Burd et al., 2018). Current research at Scotty Creek is focussed on directly measuring the flux of water through individual taliks below peat plateaus and aggregating these measurements to quantify the magnitude of this flux to the basin drainage network. The flux of water through the talik underlying seismic lines is discussed below in the section on linear disturbances.

Using CRHM, Quinton and Baltzer (2013) found that runoff from a rapidly thawing plateau decreased by 47% over a nine year period (2002-2010). The primary cause of decreased runoff was due to a reduction in the surface area of the plateau (i.e. runoff contributing area). The decrease in plateau surface area results in a concomitant increase to the surface area of the receiving wetland. Therefore, the precipitation input that would have previously been delivered to that wetland via subsurface flow through the plateau, is now delivered to the wetland directly. Where the above study examined the impact of permafrost thaw on runoff from a single plateau, subsequent field studies considered larger spatial scales in order to improve the understanding of permafrost thaw on the basin hydrograph. For example, Connon et al. (2014) investigated how

permafrost thaw affects the routing of moisture between wetlands and channel fens. This study found that as permafrost thaws, wetlands become more interconnected and therefore more capable of exchanging surface and subsurface waters, thereby increasing the runoff contributing area of drainage basins.

Haynes et al. (2018) analysed the water levels recorded at Scotty Creek for the period 2003 to 2017, and demonstrated a consistent, year to year reduction in water storage in all wetlands except for those that are hydrologically isolated from the basin drainage network. The increasing hydrological connectivity of the Scotty Creek drainage basin therefore appears to coincide with dewatering of its wetlands (Figure 8). This study evokes an image of the Scotty Creek landscape composed of peat plateaus that function as permafrost dams, and wetlands that are impounded by the latter and therefore unable to

contribute to the basin hydrograph. The authors demonstrated that the contributions to the basin hydrograph of water from the melt of ice as the permafrost dams thawed and from the resulting drainage of the impounded wetlands were both relatively minor, and that the elevated basin discharge between 1996 and 2012 was largely driven by the permafrost thaw-induced expansion of the runoff contributing area (Figure 8). Since the amount of permafrost and number of permafrost-impounded wetlands in a basin are finite, the hydrological contributions to the basin hydrograph from ice melt and from

wetland drainage are therefore transient. By contrast, the expansion of the runoff contributing area results in a permanent change to the hydrological functioning of a basin that enables it to generate more runoff per unit input of precipitation. This finding is supported by Connon et al. (2018) who demonstrated significant increases in the average runoff ratio of Scotty Creek and the other gauged drainage basins in the lower Liard River valley.

The conceptual model of preferential ground thaw and permafrost degradation proposed by Quinton et al. (2009a) provides some insight into the thaw processes that ultimately drive the expansion of the runoff contributing area. They suggested that canopy thinning due to disease, fire or other disturbance increases radiation loading to the ground surface, which leads to local thaw depressions toward which subsurface water drains. This process produces local areas of elevated soil moisture content with a concomitant increased bulk thermal conductivity. More thermal energy would then be transferred into the

ground, further deepening and broadening the thaw depression, leading to ground surface saturation, loss of tree canopy, more energy loading at the ground surface, and eventually through this sequence of positive feedbacks, lead to a local loss of permafrost. Depending upon the hydrological setting, this process could remove a permafrost dam, dewater a wetland previously impounded by the dam, and expand the runoff contributing area by an amount equal to the area of the dewatering wetland and its catchment. It is clear that the hydrological functioning of systems such as those at Scotty Creek are

deceptively complex, likely much more so than they would be under colder continuous permafrost conditions or warmer permafrost-free conditions.

The recognition of on-going permafrost thaw and resulting land-cover change forced a reconsideration of the conceptual model of basin runoff earlier envisioned by Quinton et al. (2003) and Hayashi et al. (2004), which assumed that the plateaus and wetlands were static features. Rising flows from Scotty Creek and all other gauged basins in the lower Liard River valley is often attributed to 'reactivation' of groundwater systems (St. Jacques and Sauchyn, 2009), however the low hydraulic conductivity of the glacial sediments and the minor winter-period flows of rivers in this region (<5% of annual flow) suggests other causes (WSC, 2017). Permafrost thaw-induced change to basin flow and storage processes offers a more plausible explanation (Connon et al., 2014; Haynes et al., 2018). In light of the new understanding of permafrost thaw, the analysis of how basin runoff varies with land cover type (Quinton et al., 2003) was expanded by Connon et al. (2014) for the period 1996 to 2012 and the results were re-evaluated in the context of the hydrological impacts of permafrost thaw-induced changes to the land cover. The concept that thawing permafrost produces a landscape with a higher percent cover of collapsed wetlands suggest that permafrost thaw would increase the amount of water stored in basins. However, Connon et al. (2014) found that the elevated basin discharge between 1996 and 2012 occurred in the absence of increasing precipitation over the same period (Figure 9a), and concluded that permafrost thaw should reduce the water storage of basins and that the hydrological function of collapsed wetlands should be re-evaluated for environments with widespread permafrost thaw. The finding that precipitation has not increased significantly over recent decades at Fort Simpson is contrary to what is shown by the adjusted precipitation data for the same station published by the Meteorological Service of Canada (MSC). For example, the MSC data indicates an erroneous trend of increasing winter precipitation between 1994 and 2013, the period of record coinciding with the use of acoustic sensors to measure snow depth at the Fort Simpson and other stations (Connon et al., 2018b). The magnitude of SWE is expected to be larger than the unadjusted data, but their trends should not differ. The unadjusted data also shows greater correspondence with satellite-derived data (Globsnow) and with unadjusted gauge measurements at Scotty Creek (Figure 9b).

Connon et al. (2014) also considered the hydrological impact of the thawing of permafrost barriers (i.e. peat plateaus) that separate collapsed wetlands from channel fens. This process that the authors referred to as '*bog capture*' (hereafter "*wetland capture*") transforms internally drained wetlands to 'open wetlands' that are hydrologically connected to fens. Wetland capture increases basin runoff by adding to it the runoff from direct precipitation onto captured wetlands, and the runoff entering such wetlands from their local contributing areas. As captured wetlands expand due to permafrost thaw at their margins, they merge into other collapsed wetlands, a process that further expands the basin runoff contributing area, and therefore basin runoff as well. Although Quinton et al. (2003) demonstrated that greater coverage of collapsed wetlands is associated with lower basin runoff, the hydrological function of wetlands once 'captured' is similar that of channel fens as envisioned by Quinton et al. (2003), since neither fens nor captured wetlands are hydrologically isolated from the basin drainage network. As shown in Connon et al. (2014), basins with a greater initial coverage of isolated collapsed wetlands have greater potential to exhibit a trend of increasing basin runoff as the process of wetland capture manifests over the

landscape. However, this also suggests that the increasing runoff trends (Figure 9a) have an upper limit, as wetland capture will decline as the number of remaining isolated wetlands declines. Consequently, there is a diminishing impact of wetland capture on basin runoff as permafrost thaw progresses.

Connon et al. (2014) proposed dividing plateaus into primary and secondary runoff producing areas. The primary areas are the sloped margins of peat plateaus that drain directly and continuously into the basin drainage network (i.e. channel fens). The secondary areas deliver runoff to fens through a collapsed wetland or a cascade of such wetlands whose degree of hydrological connection with a fen varies seasonally as a function of the degree of soil thaw and moisture supply. For instance, a cascade is an ephemeral flow path that is activated only during periods of high moisture supply. The wetland
cascades allow for previously isolated wetlands to interconnect via drainage channels that cut through the intervening plateau. As such, the development of wetland cascades effectively extends the basin runoff contributing area into the interior of plateau-wetland complexes. The cascading wetlands operate in a manner similar to the 'fill-and-spill' principle, where wetlands are not capable of shedding water until their storage capacity has been filled (Connon et al., 2015), similar to hydrological systems in the Prairies (Hayashi et al., 2016) and the Canadian Shield (Spence and Woo, 2003), though driven
by somewhat different processes. Connon et al. (2015) also showed that the storage deficit of an individual wetland complex is highly dependent on the wetland-to-catchment ratio. Wetlands within a relatively large catchment yields considerably more runoff than large wetlands with relatively small catchments. Such large wetlands often function as 'gatekeepers' (see Phillips et al., 2011), preventing the transmission of water to downstream wetlands. The overall wetland-to-catchment ratio of a cascade affects the total runoff from the wetland cascade as shown in Figure 10 which contrasts the runoff response of
two adjacent cascades.

Thermokarst wetland development and permafrost aggradation has likely been cyclical in this region, with a return period for permafrost re-aggradation as short as 600 years after thaw (Zoltai, 1993). Analysis of peat cores from Scotty Creek found that the collapse scar wetlands at Scotty formed at various times over the last 1250 years (Pelletier et al., 2017), while aerial
image analysis has identified several such wetlands that have developed over the last half century (Gordon et al., 2016). Previous studies on collapsed wetlands at Scotty Creek (e.g. Quinton et al., 2003; Connon et al., 2014) assumed relatively uniform characteristics among them. However, collapsed wetlands can and likely should be further sub-classified by geochemical function and groundwater connection. Gordon et al. (2016) investigated the production of methylmercury (MeHg) along the same toposequence studied by Connon et al. (2015). The authors found that the lower three collapse scar
wetlands were more appropriately described as "minerotrophic poor fens", due to the higher pH of their pore water, and presence of graminoids and sedges. Thawing permafrost leads to the expansion of both ombrotrophic and minerotrophic wetlands, and the percent cover of each of these wetland types will have implications on future water quality. Both ombrotrophic and minerotrophic wetlands can exist in close proximity and in the same drainage cascade. Although both are devoid of permafrost, the ground surface of ombrotrophic wetlands is typically above the potentiometric surface of the local

groundwater table, and therefore these wetlands recharge groundwater systems. By contrast, the minerotrophic wetlands occupy an elevation at or below the local potentiometric surface, where relatively small but consistent upward directed hydraulic gradients maintain groundwater discharge (Christensen, 2014; Gordon et al., 2016). Gordon et al. (2016) found significantly higher MeHg concentrations in the minerotrophic wetlands than in the ombrotrophic wetlands of the same wetland cascade. Similar results are reported for wetlands in non-permafrost regions (e.g. Branfireun and Roulet, 2002; Mitchell et al., 2008), suggesting that minerotrophic wetlands may be 'hot spots' for MeHg production, a major toxin that has become more prevalent in the food chain in recent years, and is of major concern to northern communities (Laird et al., 2018; Reyes et al., 2017). Understanding the trajectory of permafrost thaw-induced land cover change, including the development of such hot spots, therefore has important implications for water quality.

The effects of land-cover change from forested peat plateaus to treeless wetlands not only have a direct effect on the basin hydrology, but also affect the partitioning of energy. Chasmer and Hopkinson (2017) predict that by as early as 2044 the Scotty Creek basin could be devoid of near-surface permafrost. Using a hypothetical scenario of a complete wetland landscape, Helbig et al. (2016) calculated a regional near-surface cooling of 3-4°C at the end of winter resulting from the increased albedo of the treeless landscape assuming that the resulting landscape resembled a current wetland system. The net effect of this hypothetical landscape on snowmelt runoff is not yet understood. End of season snow water equivalent (SWE) is projected to be lower as the snowpack in channel fens is, on average 40% lower than the forests of the plateaus (Haughton, 2018). This is due to over-winter redistribution of the snowpack and wind-blown sublimation loss. The open forest canopy does not intercept a considerable amount of snow but the presence of trees greatly reduces wind speed to retain significantly more snow than the channel fens (Haughton, 2018). The amount of snow in collapse scar wetlands is highly dependent on the fetch size. Haughton (2018) found that snowmelt at Scotty Creek is primarily driven by incoming shortwave radiation, and found that snowmelt in open areas (i.e. wetlands) occurs much earlier than in forested peat plateaus. In the subarctic Yukon River basin, Semmens et al. (2013) found that the onset of snowmelt occurrs earlier primarily due to higher moisture availability for condensation and rain-on-snow events as opposed to increased temperatures. As plateaus transition into wetlands it is expected that localised cooling of near-surface air temperatures may prolong snowmelt but decreased snow cover, increased radiative loading, and higher moisture supply to the snowpack during spring will collectively accelerate the melt process. Hydrological connectivity among wetlands is highest during the snowmelt period as large volumes of snowmelt water are routed as overland flow due to restricted infiltration rates of near-surface frozen soil. Therefore, in addition to changes in snowmelt timing, more meltwater is made available to the basin drainage network. Figure 11 presents mean annual hydrographs for two 10 year periods (1976-1985 and 2006-2015) for the Jean Marie River basin, a watershed adjacent to Scotty Creek with long term hydrometric data. This figure illustrates significant increases to basin runoff during the spring freshet where runoff ratios have doubled over the 40 year period. Figure 11 also illustrates a secondary peak present in the 2006-2015 average hydrograph, but absent from the 1976-1985 average hydrograph, suggesting a departure from the nival hydrograph regime and an increasing importance of rainfall-generated runoff.

Scotty Creek occupies a landscape in transition. The close dependence of hydrology upon the energy budget and the sensitivity of the permafrost to small perturbations in vegetative cover and radiative input, changes in the regional climate directly lead to changes in the landscape; slowly transitioning from a peat plateau-dominated to a wetland-dominated system. As shown above, this has profound hydrological implications but will also impact water quality and the regional carbon balance. These changes can be further exacerbated by the presence of linear disturbances and other human intervention.

## 6 LINEAR DISTURBANCES

Linear disturbances, including winter roads and seismic lines introduced to Scotty Creek between 1942 and 1985, present a case of preferential permafrost thaw worthy of study for two main reasons: 1) the wide occurrence of these features through the Boreal and subarctic regions raises the question of their impact on local and basin runoff processes, and 2) new knowledge on how permafrost thaw beneath such disturbances affects hydrological processes can be applied throughout these regions. Linear disturbances are the most widely occurring types of anthropogenic disturbance at Scotty Creek. The density of such disturbances (i.e. total length divided by basin area) is approximately 1 km km$^{-2}$, seven times greater than the density of the drainage network of channel fens and open channels. Where the linear disturbances (i.e. winter roads, seismic lines) traverse plateaus, the tree canopy was felled and the permafrost has thawed producing a grid of linear, permafrost-free or permafrost-degraded corridors, which allow isolated wetlands to drain, and hydrological connections among collapsed wetlands, fens and plateaus to form. Williams and Quinton (2013) demonstrated that following a disturbance to the ground surface, the primary driver of permafrost thaw is the elevated soil moisture content which increases the thermal conductivity of the peat allowing more energy to be transported to the thawing frost table or permafrost table. The initial disturbance displaces the ground surface downward where it is closer to the water table. As a result, the soil moisture content (and therefore the bulk thermal conductivity) of the peat near the ground surface increases, and preferential ground thaw is initiated. Subsurface flows are then directed to the disturbance due to the topographic depression in the permafrost table further increasing the soil moisture content and rate of ground thaw in the disturbance (Quinton et al., 2009a). Therefore, once permafrost along linear disturbances thaws, it is unlikely to regenerate (Williams et al., 2013). At Scotty Creek, the ground surface of seismic lines is approximately 1 m below that of the adjacent plateaus, giving seismic lines the appearance of broad (8-10 m wide) linear ditches. In an initial investigation of the effect of seismic lines on basin hydrology, Williams et al. (2013) found that subsurface drainage from the active layer of the adjacent plateaus accumulates in surface depressions, and that during periods of high moisture supply, water drains over the seismic line from one depression to the next according to the fill-and-spill mechanism (i.e. similar to drainage through wetland cascades). In this way, runoff from plateaus can be short-circuited to channel fens via seismic lines. This study also suggested that further thaw and subsidence along seismic lines may change their hydraulic behaviour to function in a way resembling channel fens.

To further investigate the impacts of linear disturbances on basin hydrology, a 195 m segment of a seismic line cut in 1985 was instrumented in 2012. The seismic line traverses a 90 m-wide plateau, connecting a channel fen on one side of the plateau to a collapsed wetland on the other. The seismic line was instrumented with thermistors, pressure transducers (water level recorders), net radiometers, and snow depth sensors (Braverman and Quinton, 2016). Regular measurements of snow and thaw depth were made between 2012 and 2015 along the seismic line and on adjacent plateaus. Geophysical imaging was also conducted to gain insights into the rate and pattern of permafrost thaw below the line. Braverman and Quinton (2016) demonstrated that once permafrost thaw lowers the elevation of the permafrost table below the ground surface elevation of a collapsed wetland and fen on opposite sides of a plateau, the seismic line forms a hydrological connection between the two wetland types. Braverman and Quinton (2016) also computed total annual flows along the segment of the seismic line described above and found that nearly half (45%) of the total annual water flux (overland flow and subsurface flow) along that segment occurs during winter through a talik that separates the overlying active layer and underlying permafrost table. Considering the high density of the seismic line network, the widespread development of talik below peat plateaus (e.g. Connon et al., 2018a), and that taliks conduct water throughout the year, the subsurface water flux in wetland-dominated regions of thawing, discontinuous permafrost is potentially significant and could augment the dewatering process (described above) for wetlands intersected by seismic lines.

## 7 IMPACTS OF SCOTTY CREEK RESEARCH

For each of the major peatland types of Scotty Creek, the water flow and storage processes have been identified and the hydrological function in relation to the basin water balance has been defined. We have also demonstrated how these functions are changing with permafrost thaw. Although several studies in various arctic and subarctic locations have suggested that permafrost thaw may cause stream flows to increase, they do not provide a mechanistic explanation of how permafrost thaw would have this affect. However, Scotty Creek researchers showed how the gradual removal of the relatively impermeable permafrost substrate increases the lateral transfer and surface and near-surface flows, which greatly increases the proportion of the basin that is capable of generating runoff. This provides an example of how process field studies are greatly strengthened when underpinned by an extensive, high-quality data archive that includes over twenty years of permafrost and hydrometric monitoring.

Because the dominant peatland types at Scotty Creek also predominate throughout much of the subarctic, the new scientific knowledge developed at Scotty Creek on their form and functioning has contributed significantly to a wide range of scientific fields as indicated by the large number of references to studies conducted at Scotty Creek by researchers external to Scotty Creek. For example, as of November, peer-reviewed articles based on research on Scotty Creek were cited over 1350 times in the scientific literature, predominantly in the fields of water resources, environmental sciences, meteorology and atmospheric sciences (Web of Science, 2018). The dramatic changing of biophysical land cover types from forested peat

plateaus to treeless wetlands demonstrated at Scotty Creek has since been found to occur in other northern regions following the methods used at Scotty (e.g. Chasmer et al. 2016). The increasing prevalence of taliks at Scotty Creek (Connon et al., 2018a) has led other authors to investigate the extent of taliks in other regions (e.g. Lamontagne-Hallé et al., 2018), many of whom also report an increase in the number of active flowpaths, especially during the winter months when most hydrological processes were previously assumed to be dormant (Sjöberg et al., 2016; Walvoord et al., 2012). The rates and patterns of permafrost thaw documented at Scotty Creek has also informed studies examining the cycling and storage of solutes (e.g. Olefeldt et al., 2014, Korosi et al., 2015, Tank et al., 2012) and mercury (Korosi et al., 2015; Gordon et al., 2016) in the thawing landscape, as well as the cumulative impacts of permafrost thaw on aquatic ecosystems (e.g. Burke et al., 2018; Coleman et al., 2015). The rapid rate of permafrost thaw reported at Scotty led to investigations on how permafrost thaw affects aqueous (e.g. Olefeldt et al., 2014) and atmospheric (e.g. Helbig et al., 2017a; 2017b) carbon fluxes both in the Taiga Plains eco-region, and in the larger circum-polar region (Olefeldt et al., 2016).

The long-term (i.e. multi-decadal) research at Scotty Creek has greatly improved the understanding and numerical representation of the key physical, hydraulic and thermal properties of peat that blankets Scotty Creek and much of the Boreal, taiga and tundra land covers throughout the circum-polar region, has generated new knowledge and predictive capacity on the hydrological functioning of major, widely-occurring land cover types, and has provided a strong foundation to quantify rates of change, predict change trajectories, and evaluate hydrological feedbacks and outcomes. Collectively, Scotty Creek researchers have over the last few decades filled a significant gap of knowledge concerning the hydrological functioning of wetland-dominated terrains with thawing, discontinuous permafrost. Scotty Creek research has therefore complemented the new knowledge generated by researchers at other long-term stations representing other widely-occurring land-covers of the circumpolar region, including the Abisco Scientific Research Station (lowland coniferous forests to alpine); Churchill Northern Studies Centre, Manitoba (coastal subarctic); Devon Island Research Station (high arctic); Lena River Research Station (Arctic delta); McGill Subarctic Research Station / Schefferville, Quebec (Canadian Shield); Trail Valley Creek, NWT and Kuparuk River Watershed, Alaska (arctic tundra); Wolf Creek Research Basin, Yukon (subalpine), and others.

## 8 ON-GOING AND FUTURE STUDIES

The present synthesis integrates decades of insights on physical, thermal and hydrological properties, processes, pathways and feedbacks at a range of time and space scales. These insights have collectively shaped our understanding of the hydrological behaviour of the study region. As such, this synthesis also provides the wherewithal to evaluate the hydrological impacts of permafrost thaw induced land cover change. The following discussion examines the trajectory of land cover change and resulting hydrological change at Scotty Creek and the surrounding region, and in so doing, provides insights into knowledge gaps and requirements for future studies.

Carpino et al. (2018) examined the rates and patterns of permafrost thaw within twelve 36 km$^2$ areas of interest (AOIs) located along a 200 km transect extending from Scotty Creek southward to the border with British Columbia. This study helped to identify three general stages of permafrost thaw as evidenced by land cover characteristics. Interestingly, examples of each stage can be found in local areas of the Scotty Creek watershed, and as such Scotty Creek serves as a microcosm of the change observed over the larger area covered by the transect. Over decades of permafrost thaw, a plateau transforms through three general stages (Figure 2). The stage A land cover represents an early stage where collapsed wetlands are mostly hydrologically-isolated, and as such, drainage into the fen is supplied only by primary runoff (Connon et al., 2014) from the margins of the adjacent plateaus. Stage B represents an intermediate stage where primary runoff is augmented by secondary runoff from wetland cascades. The activation of secondary runoff in stage B results from hydrological connectivity among the wetlands which is absent or poorly developed in stage A. As a result, a greater proportion of the snowmelt and rainfall arriving on the land cover in stage B is converted to runoff than for a stage A land cover. Because stage B is transitional between A and C, some collapsed wetlands are hydrologically connected, while others remain hydrologically isolated. The stage C land cover represents an advanced stage of permafrost thaw, where the shrinking plateaus occur as "islands" within an expansive wetland. By this stage, plateaus are only a few tens of metres wide and therefore contain no secondary runoff and no interior collapsed wetlands. The rates of transformation between stages and how rates vary with latitude and over time remain unclear and require further study. Further study of surface and subsurface flow between ephemerally connected wetlands is also needed so that secondary flowpath contributions can be confidently predicted from numerical tools.

As the land-cover transitions through each stage, the processes and pathways governing runoff generation also transition, and as a result the three stages vary in terms of their runoff rates and patterns (Figure 12). For example, the plateaus of each stage possess a primary runoff producing area which routes water directly to an adjacent fen, but stage C has no secondary runoff producing area. Water arriving directly into collapsed wetlands or their catchments is prevented from reaching a fen in stage A, but can reach a fen in stage B if the receiving wetland is part of a cascade that is hydrologically connected to a downstream wetland or wetlands. Such activation of secondary runoff between stages A to B therefore increases runoff to fens. Primary runoff would also increase between these two stages since the fragmentation of plateaus would increase the overall length of the plateau-fen edge. In stage C, plateaus do not generate secondary runoff, so the water they receive is neither stored nor routed through wetland cascades. The runoff per unit plateau area is greatest in stage C, but because total plateau area by this stage is so low, it also has the lowest total plateau runoff. Further research is still needed to develop remote sensing methods of detecting flow connections between wetlands and of delineating wetland watershed boundaries.

By substituting space for time, the land-cover characteristics near the southern end of the north-south transect suggests the continued fragmentation and eventual disappearance of peat plateaus from Scotty Creek. The concomitant expansion of wetlands and shrinkage of peat plateaus will lead to a wetter land-cover characterised by expansive wetlands with little forest

cover. Although by stage C the hydrological connectivity would be high, the reduction of the plateau area lessens the impact of their relatively rapid flowpaths, and as a result, stage C would produce less runoff than presently observed at Scotty Creek. However, the extensive cover of black spruce forest in the southern AOIs near the border with British Columbia suggests that the wetland-dominated condition of stage C is not a final stage, but a transitional one that over time will be replaced by a forested, permafrost-free terrain. The rate of such reforestation is not well understood and requires further study.

Although Figure 12 provides a framework of permafrost thaw induced land cover, there are several areas requiring further study. For example, water flow and storage processes through the channel fens and lakes that compose the basin drainage network requires further study. There is also a dearth of knowledge on how such processes might change with on-going permafrost thaw. Winter processes such as overflow from lakes are known to be changing, however, the hydrological impact of such a change is poorly understood. Further study is also need to improve the understanding of and ability to predict how permafrost thaw affects the hydrological interaction between channel fens and adjacent peat plateaus. For example, plateaus obstruct flow causing water to follow tortuous flowpaths toward the basin outlet (e.g. see channel fens in Figure 2). During period of high flow through fens, water is abstracted into peat plateaus and then released following the high flow period. Because plateaus shed most of the water they receive, their presence on the landscape displaces upward the water table in the intervening wetlands, increasing the frequency of overland flow and rapid flow through the highly conductive near-surface peat. These obstruction, abstraction and water table displacement functions of plateaus are not well studied, and as a result, how these functions might change with permafrost thaw cannot be predicted with confidence.

Recent field studies at Scotty Creek have identified the growing presence of "treed wetlands". From aerial imagery, these features are easily mistaken for peat plateaus owing to their tree-cover; however, they are wetlands and as such contain no permafrost. The development of treed wetlands at Scotty Creek sheds light on how treeless wetlands can transition back to forest, as observed in the southern AOIs. The ground surface topography of collapsed wetlands that are hydrologically-isolated is characteristically flat. However, the dewatering of such wetlands as a result of "wetland capture" or their intersection by a seismic line appears to be coupled with development of a hummocky micro-topography. In actively dewatering wetlands, hummocky micro-topography is most pronounced on the sloping wetland margins where drainage would have occurred first. According to Zoltai (1993), hummocks play an important role in the regeneration of permafrost since their vertical growth enables the development of multi-year "ice bulbs" resulting in the evolution of hummocks into embryonic, and ultimately mature plateaus before collapsing back into the wetland form. In the present warming climate, it is doubtful that hummocks can complete this evolutionary cycle. However the establishment of a black spruce forest does not require permafrost, but the dry near-surface conditions provided by peat plateaus and hummocks. The black spruce stands of treed wetlands differ from the stands on adjacent plateaus is several ways. For example, in treed wetlands, the trees are clustered onto hummocks and the average age of the tree stand is significantly lower than on the plateaus. The reason for

the latter is because trees eventually grow to a size and weight too great for a hummock to support, resulting in hummock collapse and waterlogging of the tree roots and eventually tree mortality. The hummocks are therefore not as effective as plateaus at providing the necessary dry conditions throughout the life of a tree, however, it is expected that as the treed wetlands continue to dewater over the coming decades, much of the forest lost due to permafrost thaw will regenerate. The rates and patterns of this regeneration, the environmental processes and feedbacks affecting it, and its hydrological implications are areas requiring further study.

**Author Contributions**

All authors were involved in synthesizing the presented information and preparation of the manuscript.

**Competing Interests**

The authors declare that they have no conflict of interest.

**Acknowledgements**

We gratefully acknowledge the support of the Dehcho First Nations, in particular, the Liidlii Kue First Nation and Jean Marie River First Nation. This work was supported by the Natural Sciences and Engineering Research Council of Canada (NSERC) through their funding of Discovery Grants and a Collaborative Research and Development Grant (Consortium for Permafrost Ecosystems in Transition - CPET). We also acknowledge support for infrastructure at the Scotty Creek Research Station through the Canadian Foundation for Innovation (CFI).

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

**Table 1: Instrumentation at Scotty Creek Research Site. A map showing the locations of scientific and other infrastructure at the Scotty Creek Research Station is available at http://www.scottycreek.com/. This on-line map is regularly updated.**

| | |
|---|---|
| 1996 | Scotty Creek gauging station installed |
| 1999 | Old Camp (seasonal) established |
| | Installed: water level recorders, ground temperature sensors, hydrometric sensors |
| | Initial ground thaw surveys, snow surveys |
| 2001 | Instrumented (temperature, soil moisture) soil pits, wells, expanded hydrometric sensor arrays |
| 2003 | Climate Station (Plateau 1: degrading plateau, sparse canopy) |
| 2004 | Climate Stations (Bog 1: direct connection to channel fen; Fen 1: channel fen draining small lake; upgraded Plateau 1 station) |
| 2006 | Snow courses introduced for annual snow surveys |
| 2007 | Climate Station (Plateau 2: stable plateau, dense canopy) |
| 2008 | First Lake Camp (all-season) established |
| 2009 | Deep (10 m) thermistors installed |
| 2010 | 10 m flux tower |
| 2012 | Goose Lake Camp (all-season) established |
| | 18 m flux tower and complementary Bog flux station |
| | Initial instrumentation of seismic line: climate stations, deep thermistor profiles, experimental thermosyphons |
| 2014 | Climate Stations: Plateau 3 (stable plateau, sparse canopy), adjacent Burned Forest Station and Non-Burned Station |
| | All water level recorders anchored and logging year-round |
| 2015 | Climate Station (Fen 2: channel fen, large drainage lake) |
| 2017 | Boardwalks and Expansion of Goose Lake Station infrastructure |
| | NASA / ABoVE flights for remote sensing image acquisition over Scotty |

**Captions for new Figures:**

Figure 1: Distribution of wetland-dominated terrain containing discontinuous permafrost throughout the southern NWT (a), including the Scotty Creek drainage basin (b).

Figure 2: A classified image of a 22 km$^2$ area of the Scotty Creek drainage basin showing the distribution of peat plateaus, isolated and connected collapsed wetlands, and channel fens. Also shown are areas where isolated collapsed wetlands (A), bog cascades (B), and plateau islands (C) predominate. An enlargement of "B" identifies the wetland cascades of two adjacent catchments.

Figure 3: Cross section of a peat plateau at Scotty Creek based on measurements of supra-permafrost thickness and ground surface elevation at 1 m intervals showing the difference in depth and lateral extent of the permafrost table between 2006 and 2015 (a) and the development of a talik in 2015 (b).

Figure 4: Depth variation of saturated horizontal hydraulic conductivity (*K*) measured from tracer (KCl−) tests conducted at Scotty Creek, as well as at other organic-covered permafrost terrains in north-western Canada, as reported by Quinton et al. (2008). The light and dark grey indicate the zones in which *K* is uniformly high and uniformly low, respectively. The schematic indicates the decrease in the mean hydraulic conductivity of the saturated flowzone with active layer thaw. Modified from Quinton et al. (2008).

Figure 5: Suprapermafrost layer thickness measurements ranked from (left) thinnest to (right) thickest and associated thaw depth measurements. Active layer refreeze depth was measured in April 2016, following thaw depth measurements in August 2015. Modified from Connon et al., 2018a.

Figure 6: Change in permafrost width and the suprapermafrost layer thickness for the period 1999-2018. Modified from Quinton et al. (2011)

Figure 7: Taliks can either be isolated systems within a plateau, or continuous features across a plateau that connects adjoining wetlands. Adapted from Connon et al. (2018).

Figure 8: Water loss from four connected wetlands at Scotty Creek for the overlapping period of the runoff and water level records (coloured lines). Total annual runoff measured at the Scotty Creek basin outlet and total annual precipitation (rain and snow) measured at Fort Simpson for 1996 to 2015. Shaded area denotes extra observed runoff beyond that expected based on the mean of runoff ratios (R.R.) prior to 1996 and the quantity of precipitation. Runoff from the neighbouring Jean-Marie River is plotted to illustrate the similar timing and magnitude elevated basin runoff. All runoff amounts are totalled over the water year (1 October–30 September). Adapted from Haynes et al. (2018).

Figure 9: Historical precipitation, temperature and runoff at Scotty (a). Runoff after 1995 was measured directly by the Water Survey of Canada, and runoff prior to 1995 was estimated by interpolation from Water Survey of Canada data at an adjacent watershed (Jean-Marie River). Precipitation and air temperature were measured at the Fort Simpson A Climate Station by Environment and Climate Change Canada. Adjusted and unadjusted total

annual snow water equivalent (SWE) at Fort Simpson Airport (b). Also shown is the SWE measured at Scotty Creek (Geonor gauge with Alter shield), and SWE provided by the Globsnow data set for the same region.

Figure 10: Hydrographs for two adjacent series of wetland cascades (East and West) during the summer of 2014. The plateau to wetland ratio of the East cascade is four times higher than that of the west cascade, explaining the differences in runoff magnitude between the two cascades. Adapted from Connon et al. (2015).

Figure 11: Composite hydrograph for two 10 year periods (1976-1985 and 2006-2015) of runoff from the Jean Marie River basin.

Figure 12: The transformation of peat plateau runoff generation processes with increasing thaw from left to right. PP = peat plateau, CF = channel fen, CW = collapsed wetland, I = isolated, C = connected. The term "shed" refers to the watershed of a land cover feature. For example, "CW shed" refers to the water shed of a collapsed wetland. Modified from Quinton et al. (2017).

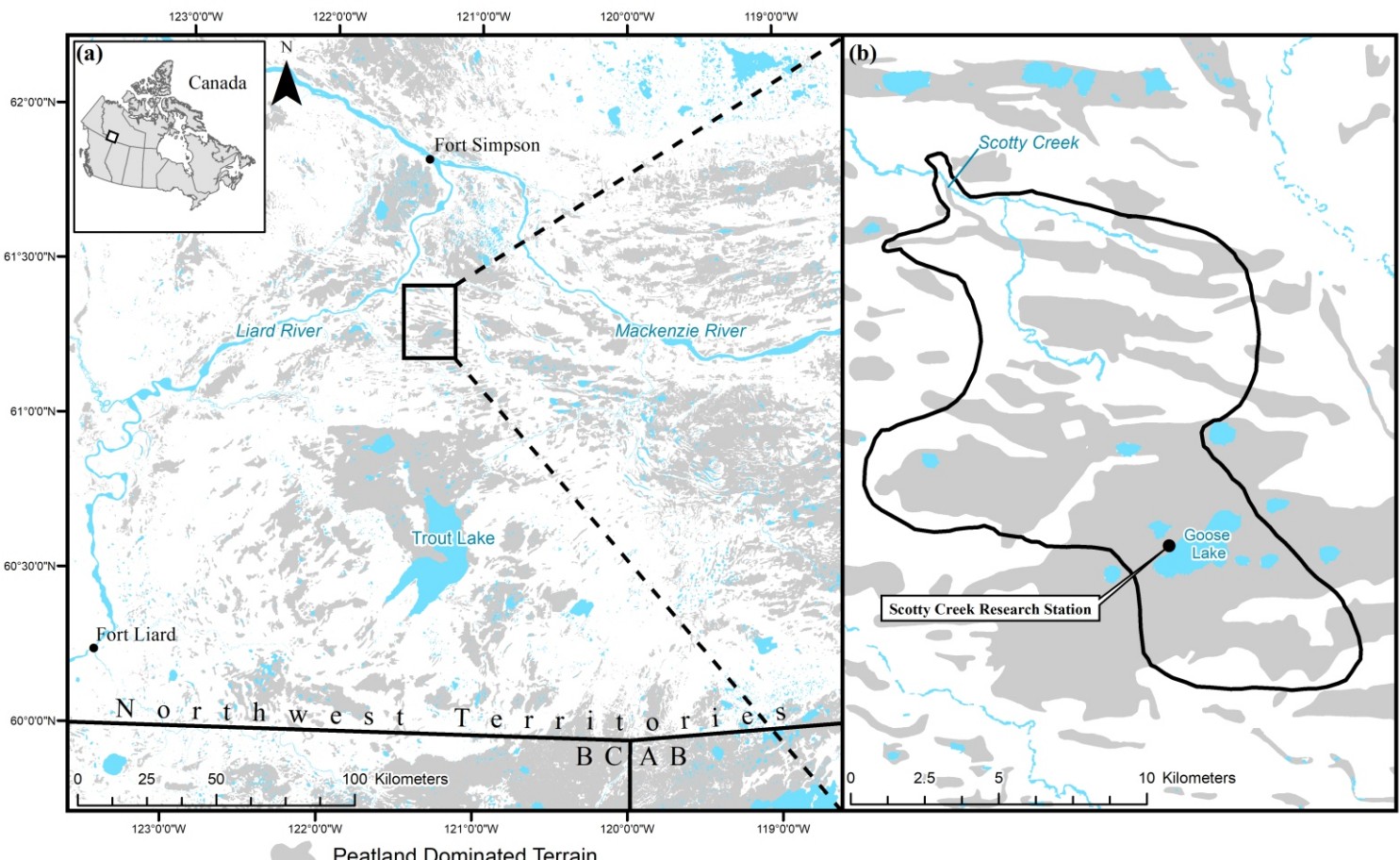

Peatland Dominated Terrain

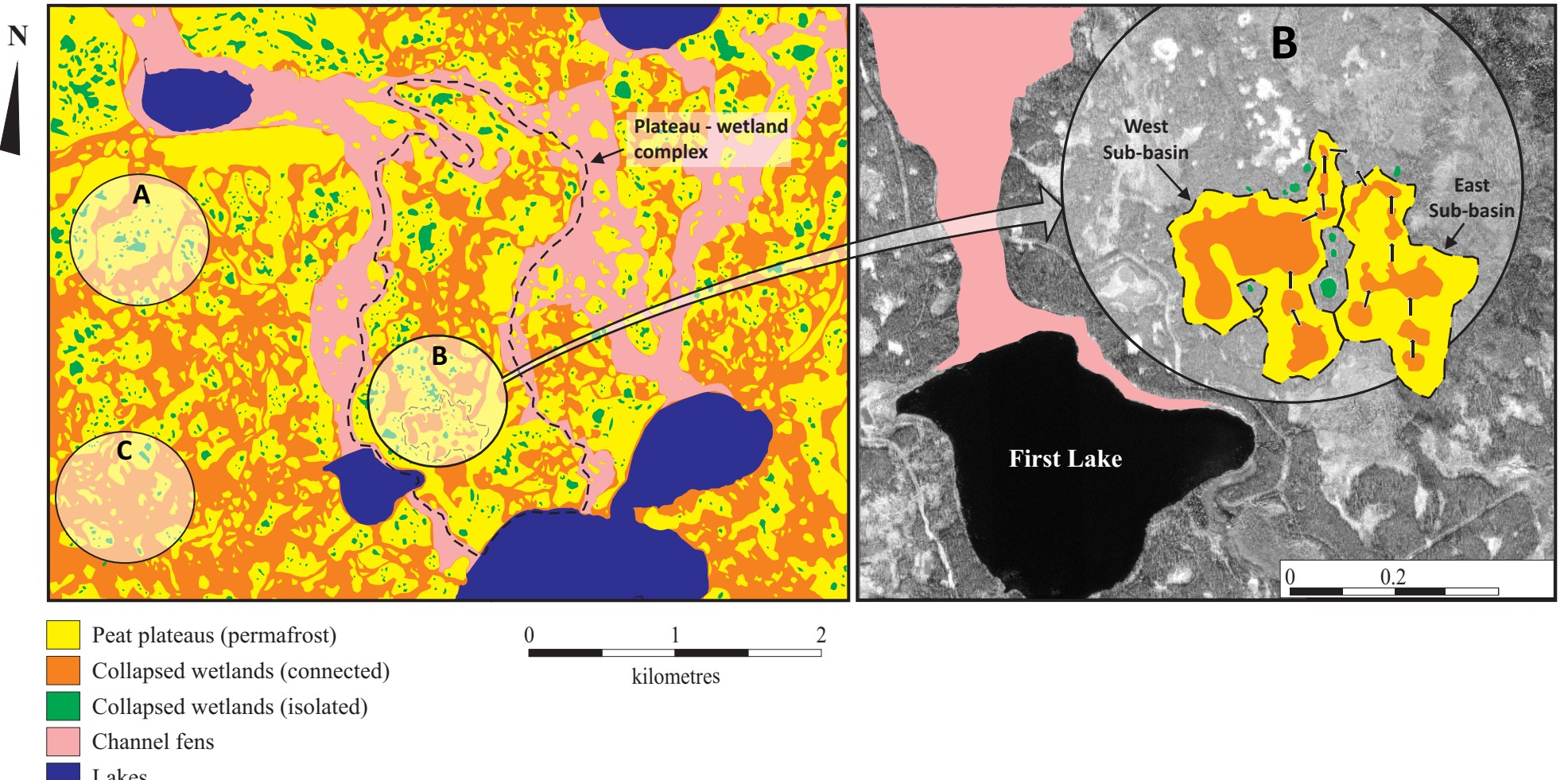

N

Peat plateaus (permafrost)
Collapsed wetlands (connected)
Collapsed wetlands (isolated)
Channel fens
Lakes

Plateau - wetland complex

A

B

C

0          1          2
kilometres

B

West Sub-basin

East Sub-basin

First Lake

0          0.2

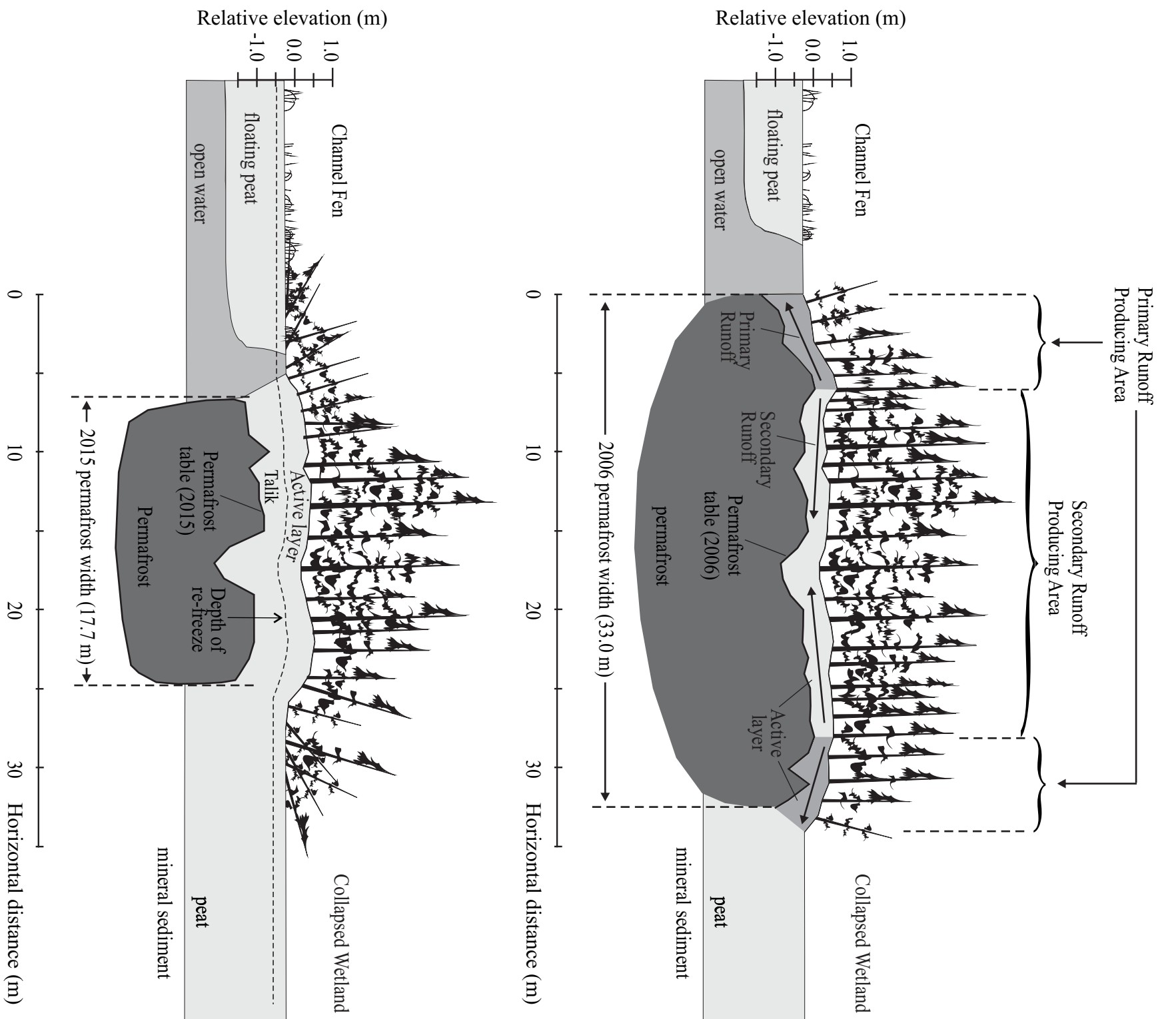

Figure 3

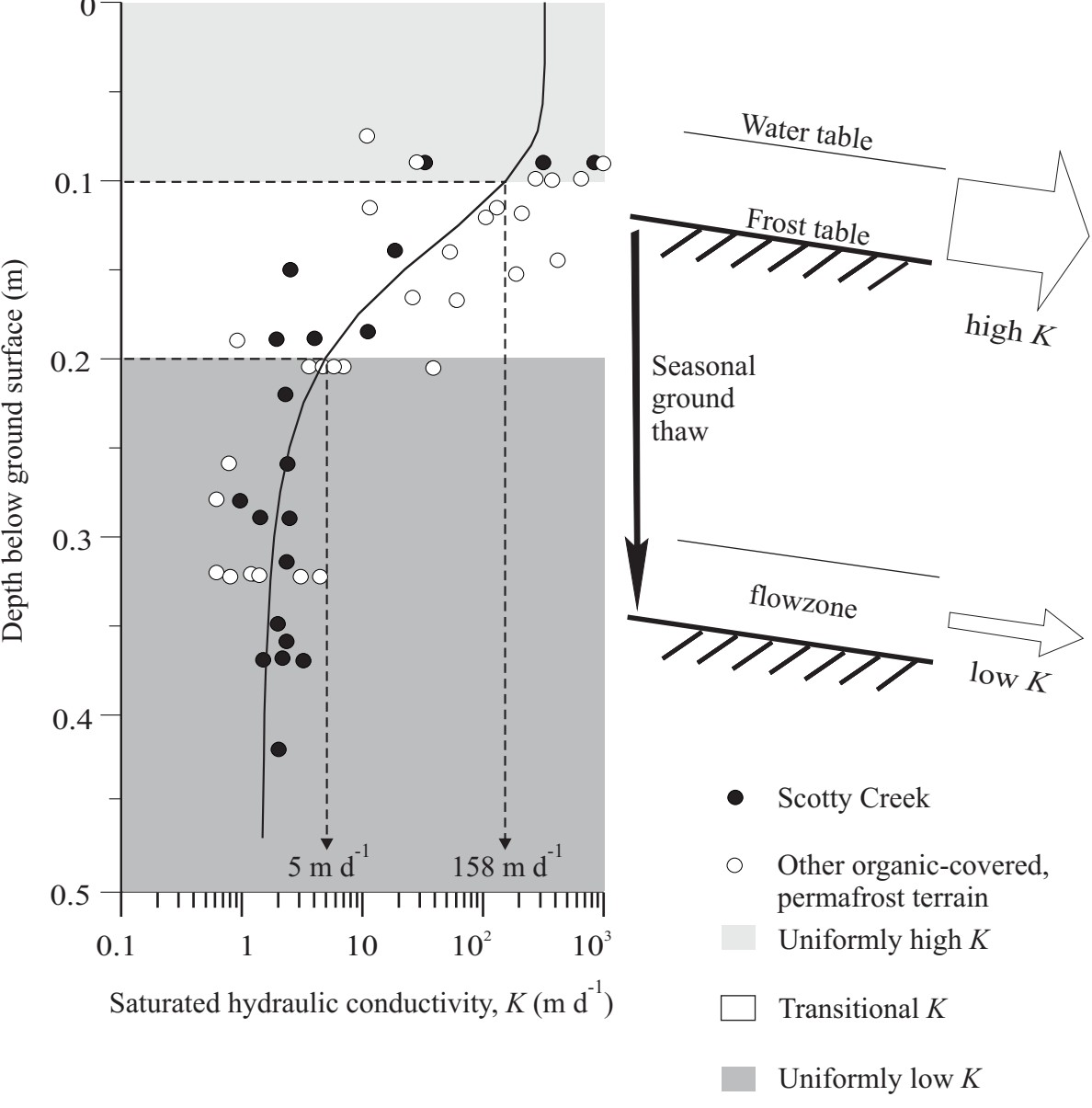

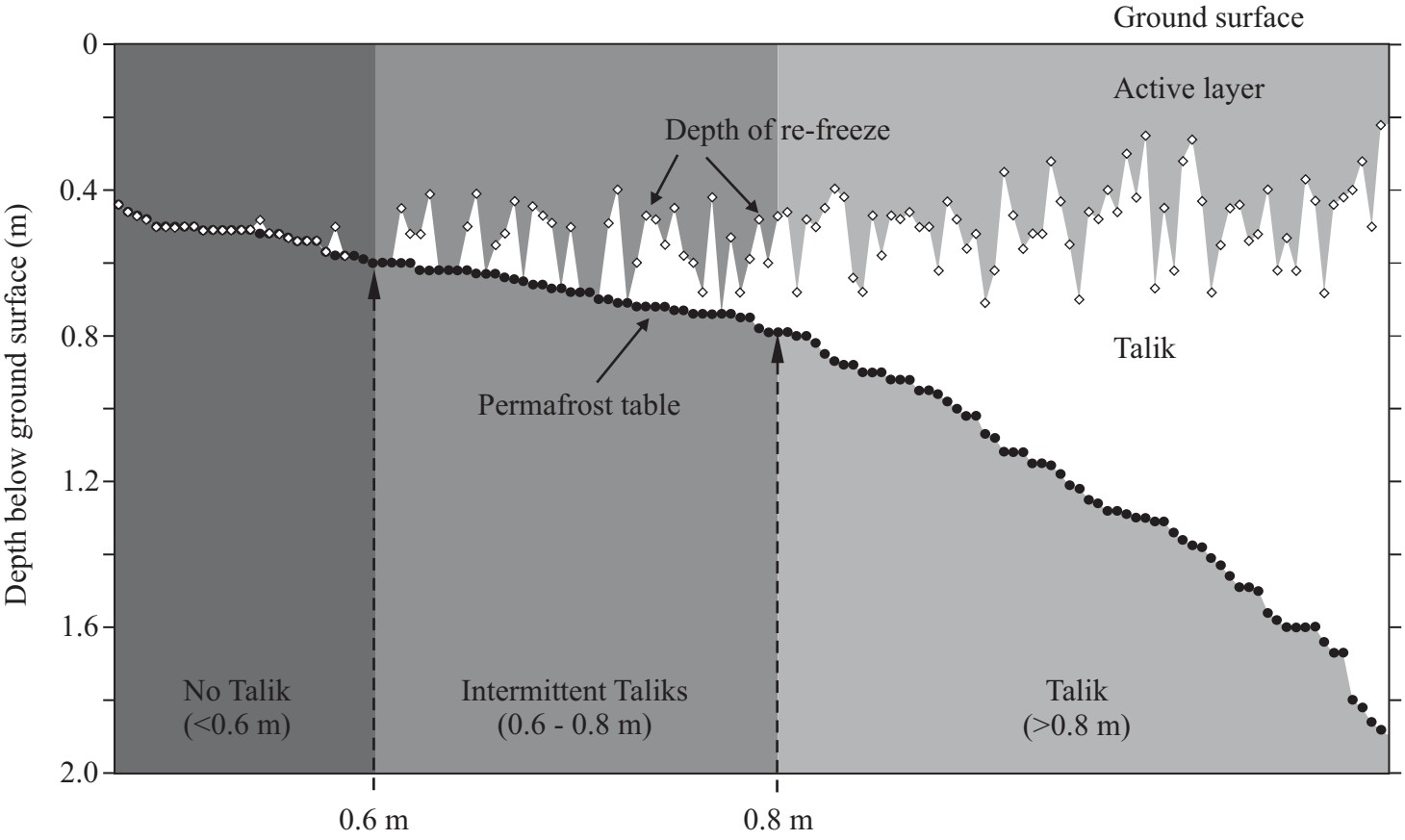

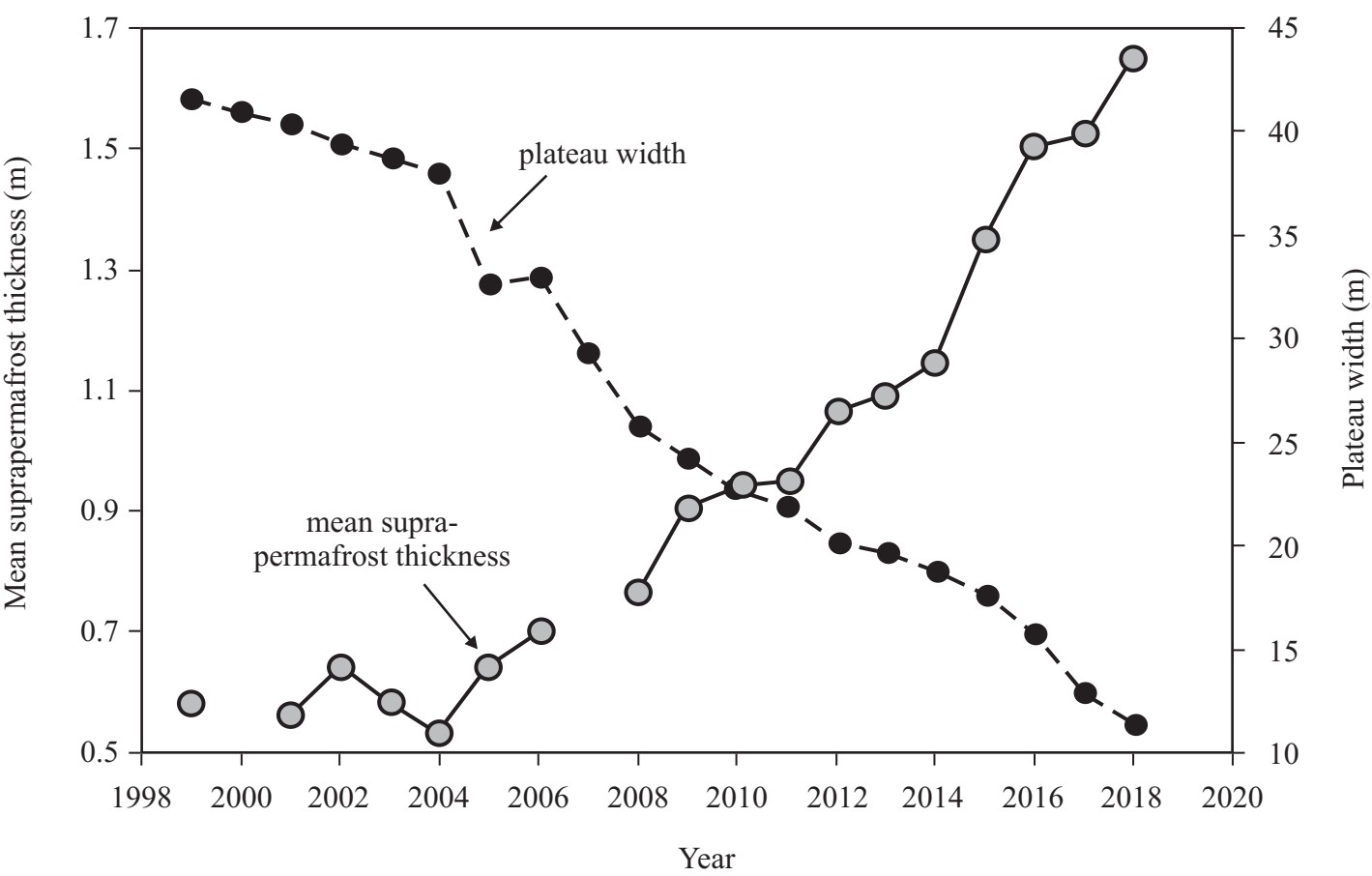

**Isolated Talik**

Active Layer

Talik

Permafrost

**Connected Talik**

Active Layer

Talik

Permafrost

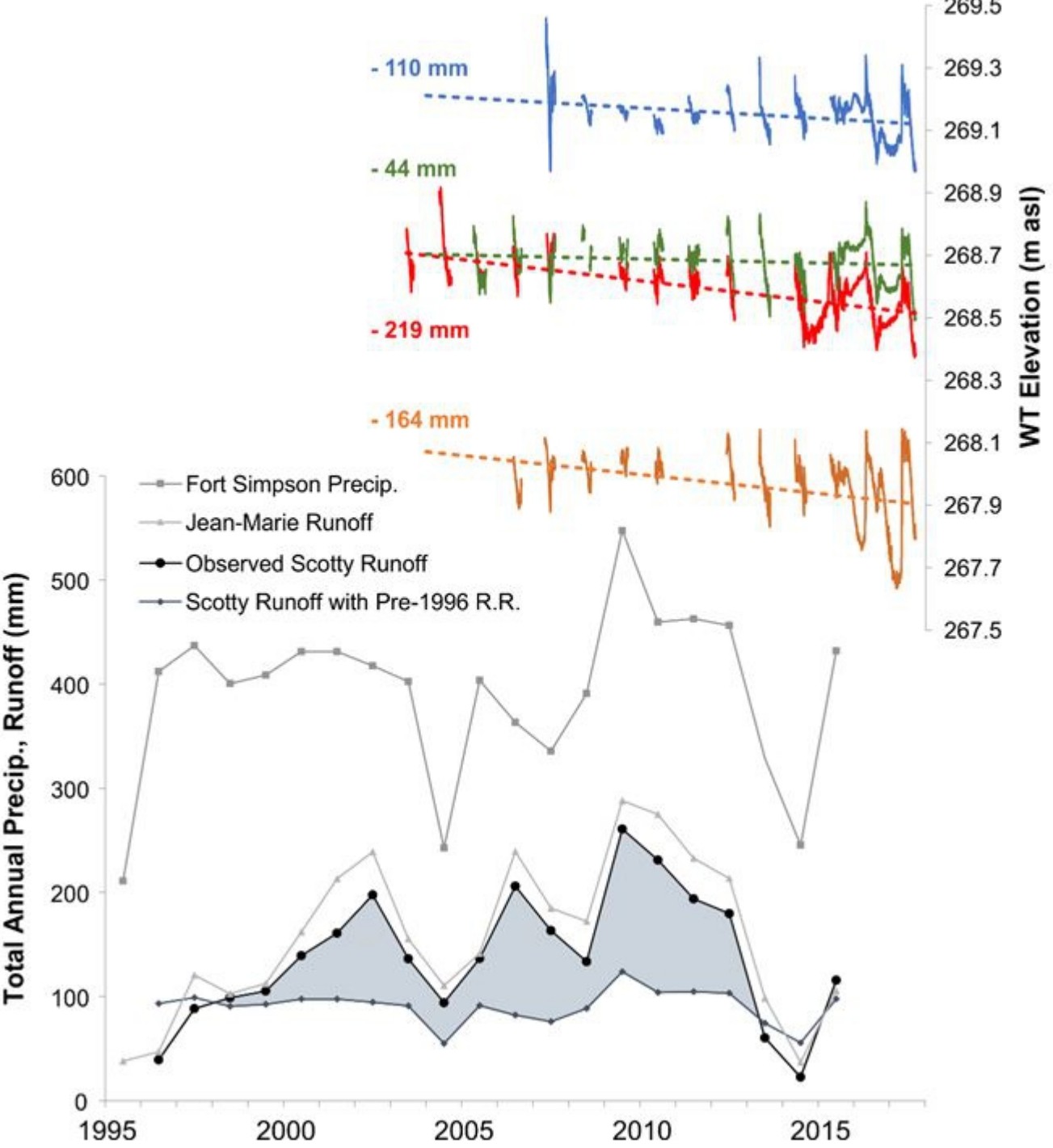

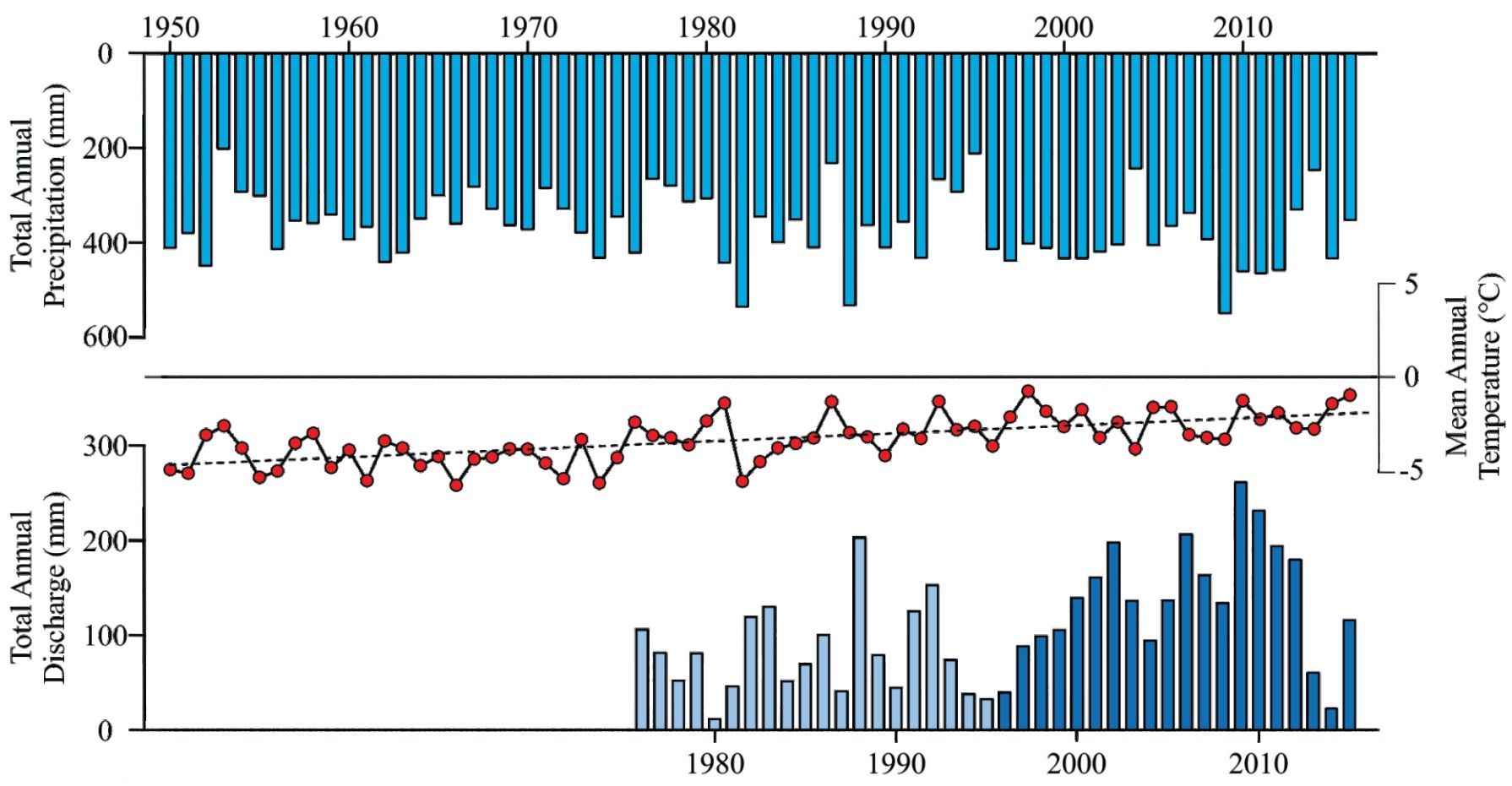

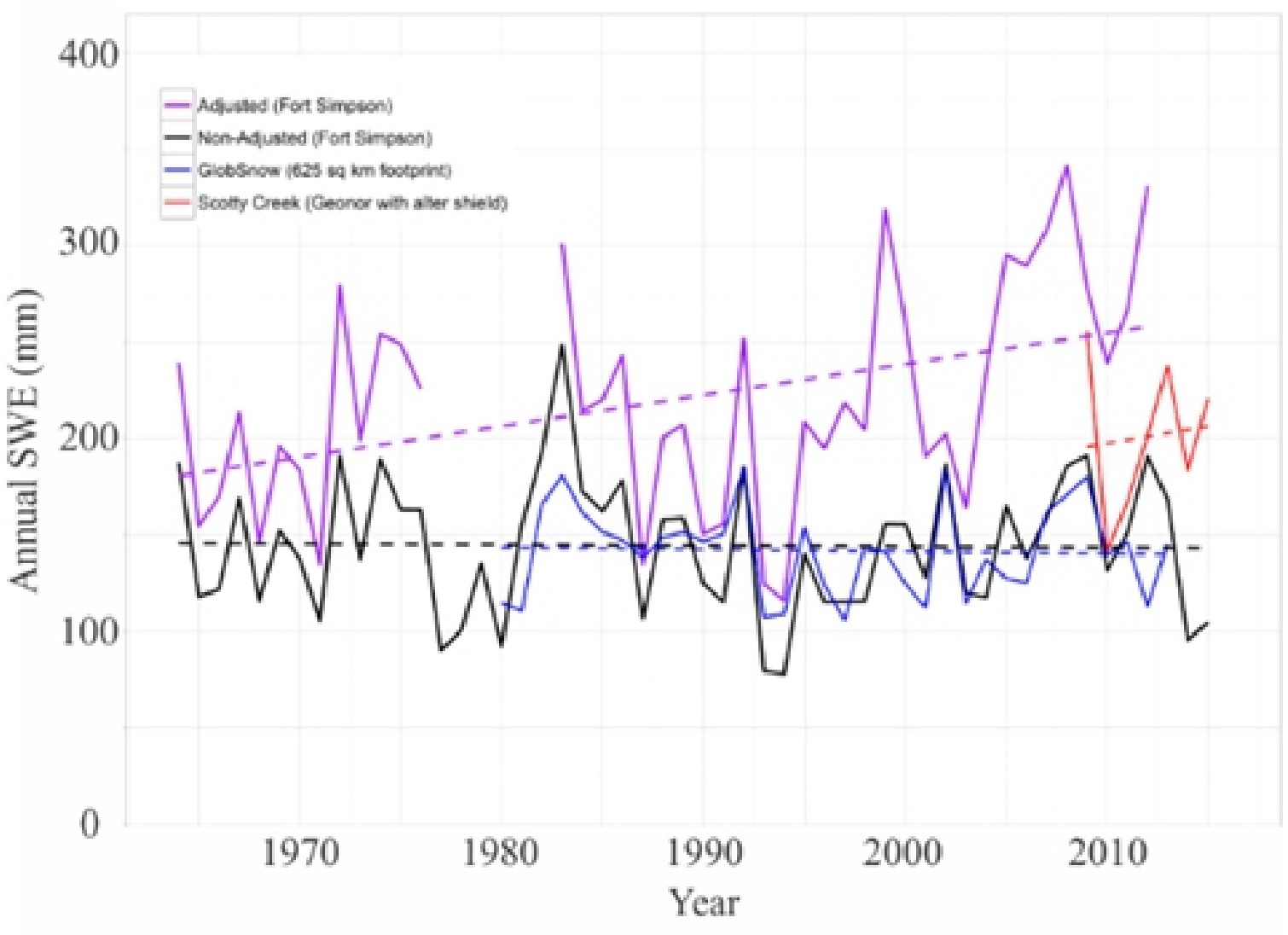

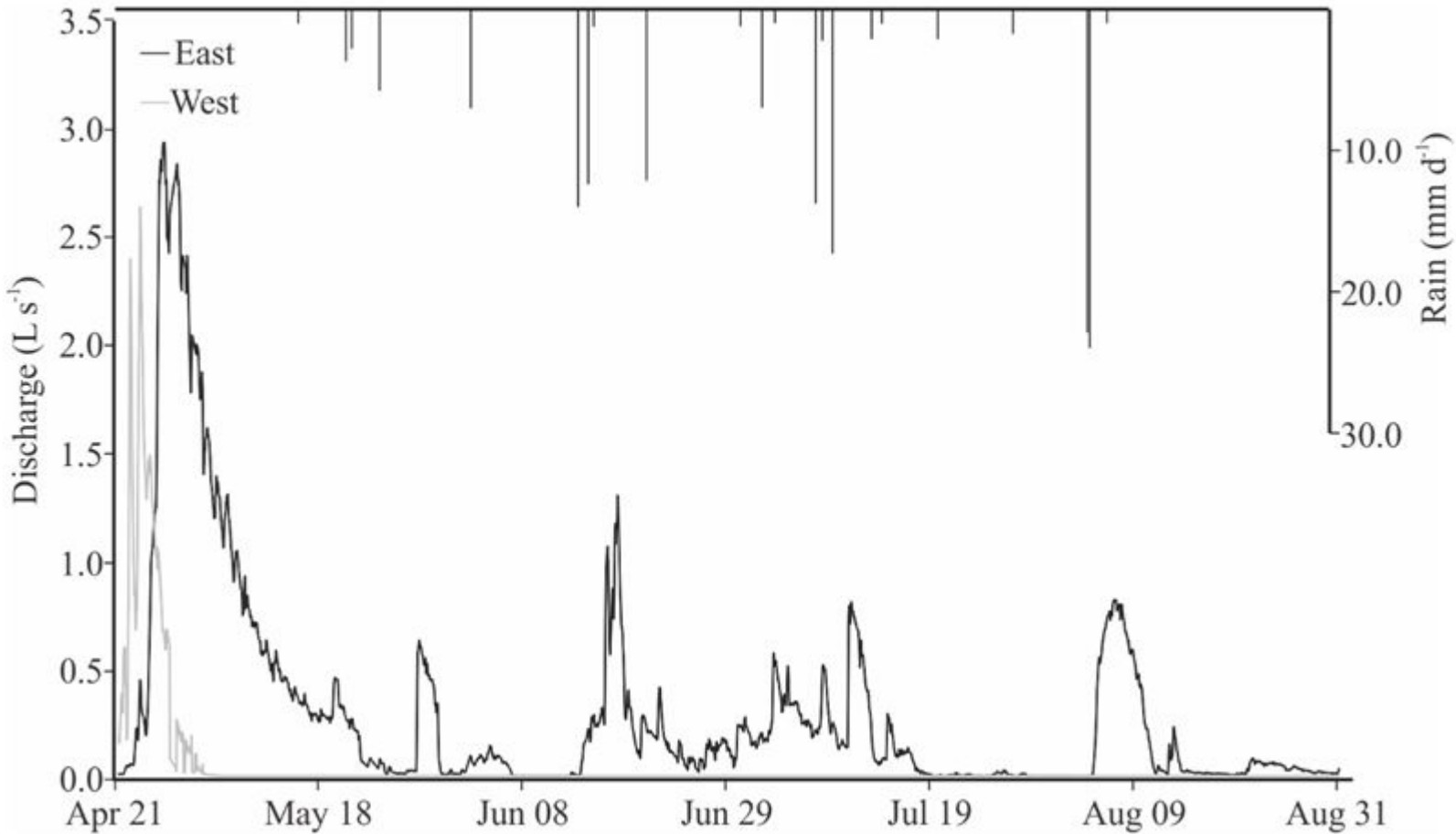

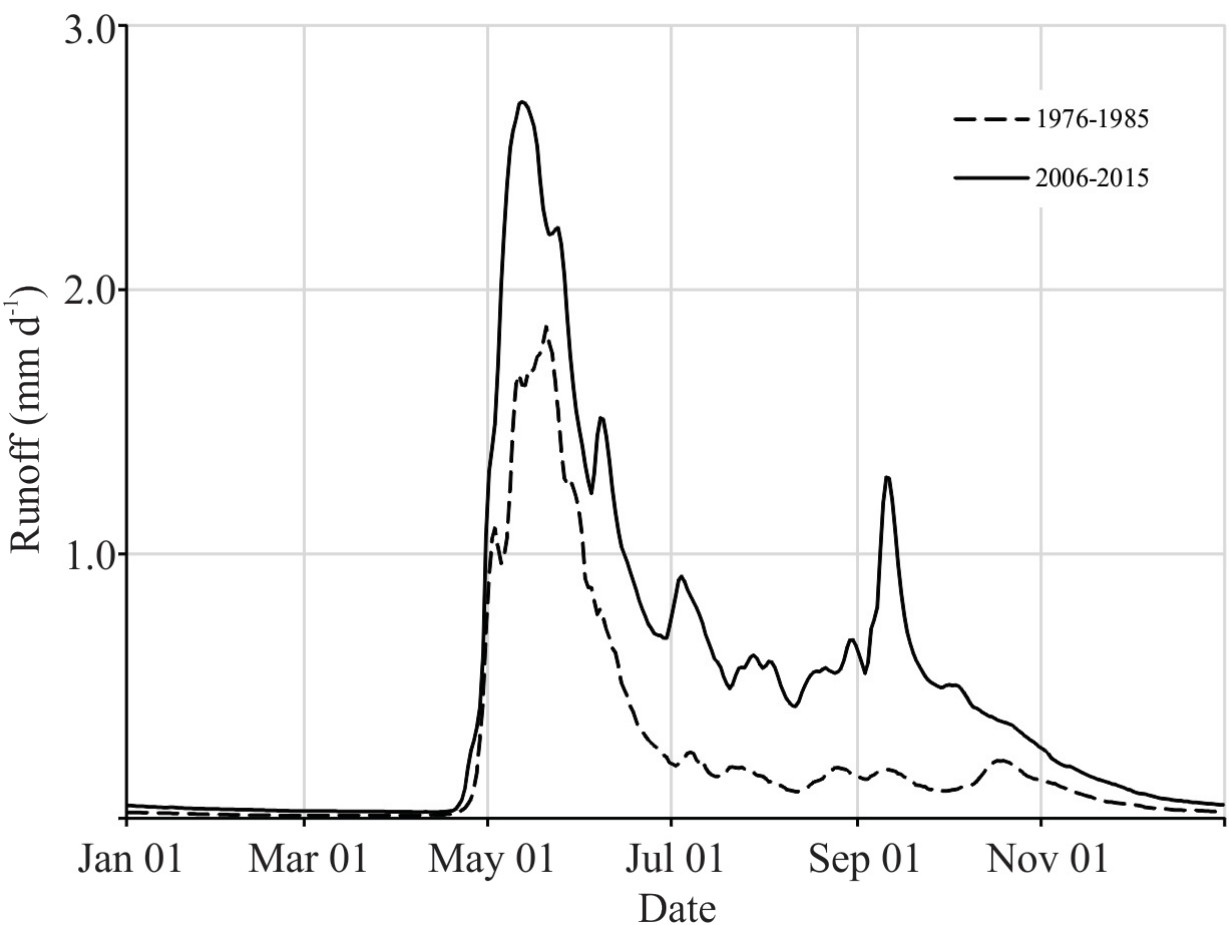

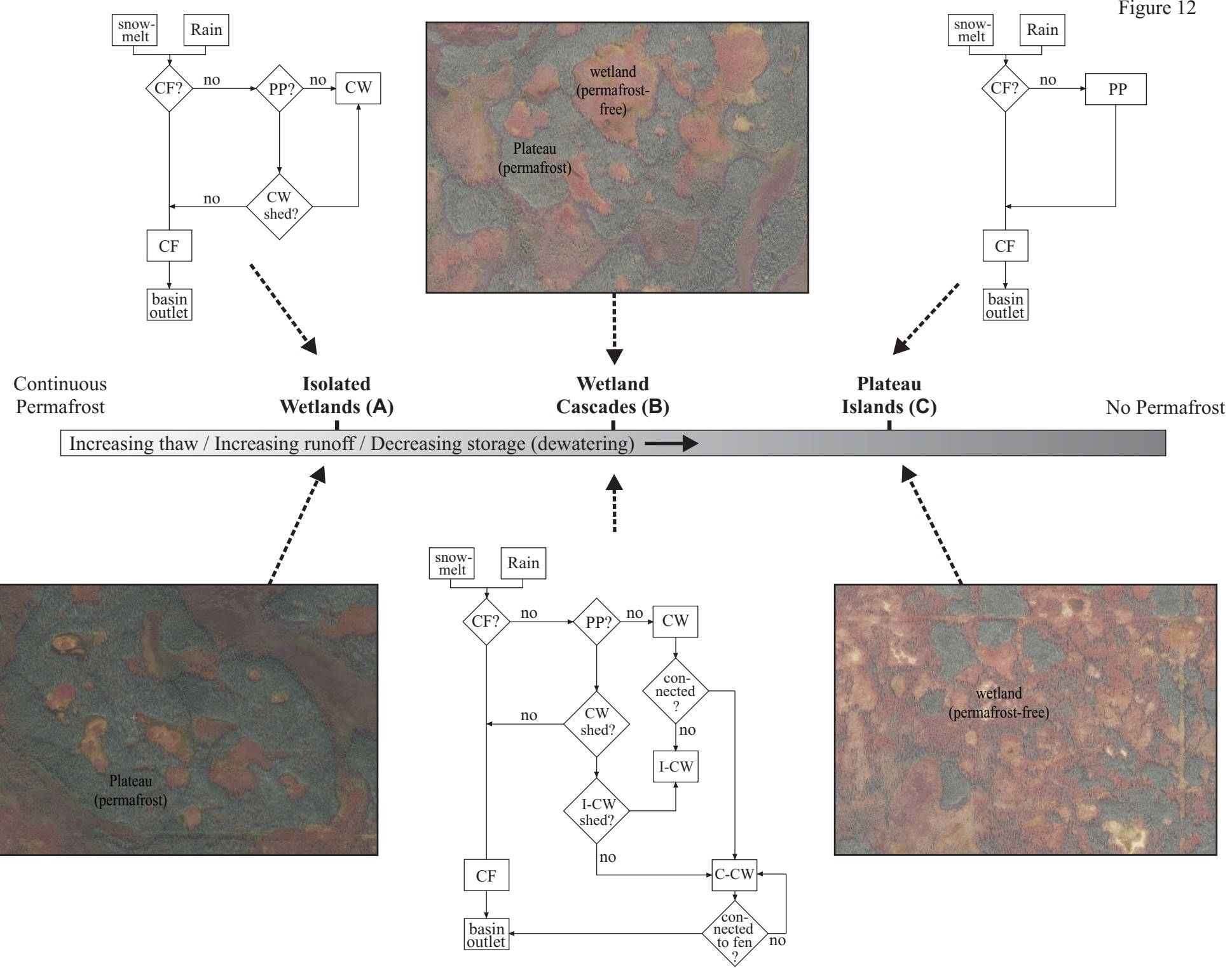

Figure 12