# Peer review of "A Synthesis of Three Decades of Hydrological Research at Scotty Creek, NWT, Canada"

_Hydrology and Earth System Sciences, 2018_

## Short Comment (SC1) · 16 Aug 2018

For appropriate context and thoroughness, I think it is important to mention a few additional Environment Canada-National Water Research Institute sponsored studies in this review. The first two papers are based on the nearby and quite similar Manners Creek watershed which preceded much of the other hydrological research in this area (beginning in 1989/90), and predated establishment of the Scotty Creek gauge:

Craig, D. 1991. Geochemical evolution of water in a continental high boreal wetland basin: preliminary results, Northern Hydrology: Selected Perspectives, T.D. Prowse and C.S.L. Ommanney (Editors), NHRI Symposium No. 6, National Hydrology Research Institute, Environment Canada, Saskatoon, Saskatchewan, pp. 47-55.

Gibson, J.J., Edwards, T.W.D., Bursey, G.G., Prowse, T.D., 1993. Estimating evaporation using stable isotopes: quantitative results and sensitivity analysis for two catchments in northern Canada, Nordic Hydrology 24, pp. 79-94.

The latter is a comparison of evaporation loss in Manners Creek and a watershed near Baker Lake.

And the following papers that include measurements in Scotty Creek and other similar basins in the area:

Gibson, J.J., Prowse, T.D., 1999. Isotopic characteristics of ice cover in a large northern river basin, Hydrological Processes 13, pp. 2537-2548.

Gibson, J.J., Prowse, T.D., 2002. Stable isotopes in river ice: identification of primary over-winter streamflow signals and their hydrological significance, Hydrological Processes 16, pp. 873-890, http://dx.doi.org/10.1002/hyp.366.

Stadnyk, T.A., St. Amour, N.A., Kouwen, N., Edwards, T.W.D., Pietroniro, A., Gibson, J.J. 2005. A groundwater separation study in boreal wetland terrain: the WATFLOOD hydrological model compared with stable isotope tracers, Isotopes in Environmental and Health Studies 41(1), pp. 49-68, http://dx.doi.org/10.1080/10256010500053730.

St. Amour, N.A., Gibson, J.J., Edwards, T.W.D., Prowse, T.D., Pietroniro, A., 2005. Isotopic time-series partitioning of streamflow components in wetland-dominated catchments, lower Liard River Basin, Northwest Territories, Canada, Hydrological Processes 19, pp. 3357-3381, http://dx.doi.org/10.1002/hyp.5975.

The isotopic and related datasets described in the latter two papers represent some of the most complete time-series ever published in Canada, and have been used in development and refinement of the isotope-enabled distributed models.

---

## Referee Comment (RC1) · Anonymous Referee #1 · 29 Aug 2018

This is in general an excellent and important paper that summarizes key insights into the functioning of discontinuous permafrost hydrological systems in western Canada, based on a 20 year field research prgramme. There are a few issues of clarity that need to be addressed; P2 Line 18 discusses precipitation and stream gauging networks with reference to figure 2. The expectation is of a map that shows gauging sites, but this is missing. P3 section 2. This is a paper about hydrological functioning, but no basic data are provided to introduce the climate context for the basin. We need a brief summary of the climatic setting and precipitation characteristics, and a sentence or so on the flow regime. P3 line 2 delete 'through this process'. It's not clear what this process is and the phrase is in any case unnecessary. P4 line 9 delete comma P4 Line 13 maybe 'primary hydraulic function' - there are many functional aspects of these systems P5 line 2 insert

comma after 2003) P5 Lines 14, 16, 17 – too many ‘unique’s – poor English. P6 line 4 ‘total porosity decreases by 10%’ - please be specific, is this a 10% decrease in the porosity value, or is the porosity expressed as a percentage changing by 10? There is a use of undefined terminology here – what is meant by water storage coefficient, and active porosity? How is active porosity different from drainable porosity? How is a decrease of ‘only 10%’ consistent with a ‘rapidly decreasing’ storage coefficient? P6 Line 18 k is loosely defined here as permeability. Presumable intrinsic permeability is what is meant. P9 line 25 retained rather than detained P10 line 5 delete comma after 2003, insert comma after ‘that’ p10 line 30 the SFASH model p12 line 20 insert comma after plateaus, In the latter case. . .. P16 line 7 what precisely is meant by a composite hydrograph? Is this the mean over the period, or something different? P16 line 28 what precisely is meant by ‘soil moisture’ here? It’s difficult to see how total moisture would increase if porosity decreases. P17 Summary Please summarize conclusions for basin storage. Only flows are discussed. P18 the paper ends quite abruptly. We need a section of reflection here. After such extensive filed work over a long period, what are the outstanding gaps in knowledge and primary requirements for future work?

---

## Referee Comment (RC2) · Anonymous Referee #2 · 30 Aug 2018

General comments

Quinton et al. compile, summarize, and relate a number of studies that have led to a better integrated knowledge of environmental change (esp. hydrologic change) in the Scotty Creek catchment, Northwest Territories. This is an important research catchment in the context of Canadian cold regions hydrologic research and is increasingly recognized as an important long-term study site for informing our knowledge of changes in water resources across the circum-polar subarctic. The article is mostly well written, and is an excellent first-person (people?) account of the work conducted at this important site. It will be of general interest to the readers of HESS, especially those focused on cold regions.

Major comments

1. My main criticism is that this article could be alternatively titled ‘My life’s work’. There are ∼40 self-citations to the first author’s work alone, which is highly unusual. However, I can appreciate that there is no real way to circumvent this, as doubtless the authors of this paper (especially the first author) have led the work at this site for many years and produced many important papers. However, as a comment, I wonder if they could include a section (or at least some more text) relating the work at Scotty Creek to other long-term northern, discontinuous permafrost research catchments, even ones that may have different physical conditions. That might give this broader appeal.

2. I don’t find the first two paragraphs of the introduction particularly well written. In my opinion the first paragraph should talk about circum-polar subarctic cold regions/permafrost/hydrology work to begin with (at a large-scale) and then narrow to the region containing Scotty Creek. This paper starts with a rather abrupt introduction to the Liard River valley, and then expands in the second paragraph to the broader Mackenzie River basin. Restructuring and enhancing the circum-polar questions and implications would position this paper better in the broader literature. I think that is important, as it is certainly not standard to write a review paper in a hydrology journal that is focused on just one site, and it would be useful for the authors to really emphasize the critical nature of this work.

3. Sections 2 and 3 are a bit of a laborious read and describe processes and concepts that could easily be visually represented. Why not have a figure in Section 2 that shows the landscape units and how they function and interact hydrologically? This would be useful when the authors refer to the ‘new conceptual framework for runoff generation’ (P5, L23). Some of this would be easier to see than read (although I recommend that the text be retained). Section 3 could have a figure that shows some active layer conditions or processes (e.g. data for K vs. depth or some freeze-thaw conditions).

4. I miss much discussion on biogeochemical processes or storage and lateral fluxes of nutrients etc. I understand that has not been the focus at Scotty Creek, but I am surprised there are so few studies to refer to. Related to this, there is very little discussion on vegetation changes at Scotty Creek except where the focus is the impacts on hydrologic processes. Surely some of the Baltzer work could be better highlighted. I would suggest giving more than a passing glance to the important Baltzer et al. 2014 GCB (P11, L26) study. Although Eco-Hydrological is included in the title, there surely isn't much eco content in the paper.

5. Figure 3 on the active layer and talik etc. could also include an additional pane or two showing the 'dual layered system' of flow (P12, L14).

6. How much longer will the increased hydrologic connectivity result in higher flows? If the wetlands are dewatering, when will the basin reach 'peak water'?? It seems to me that this concept related to glacier discharge could also be applied to Scotty Creek as the wetland storage decreases.

7. The conclusion ends abruptly. It almost seems like text were deleted by accident. There is no vision, no path forward, no high-level statement explaining the importance of the work. Something needs to be rewritten here.

Minor comments

P2, L5, should 'regions' be after 'warming'? Figure 1, this is obvious to Canadians, but probably 'Canada' could be labeled in the inset Way et al. (2018) report a study of isolated permafrost patches that I think could be tied to the work at Scotty Creek. https://www.the-cryosphere.net/12/2667/2018/

---

## Referee Comment (RC3) · Anonymous Referee #3 · 1 Oct 2018

Review of journal article titled: "A Synthesis of Three Decades of Eco-Hydrological Research at Scotty Creek, NWT, Canada" by Quinton et al.

General comments:

This study provides a comprehensive synthesis of research produced at the Scotty Creek Research site. It is well-framed and provides an interesting and useful resource for an audience interested in processes and modelling in cold wetland-dominated regions. I think that the paper is well-written, although does occasionally come across as a series of paper summaries. As this is a review-type of article, I do not find any major flaws within the paper, and think that it can be published after addressing several mostly minor comments.

Overall, I recommend minor revisions, as my comments are minor and easily addressed with some effort. This is an excellent demonstration of the need for and benefits of long-term research locations, and it should be promoted as such. Below are my specific comments.

Specific comments:

Page 2 Line 18: The reference to Figure 2 here is odd, since the figure does not indicate anything about expanding precipitation and stream gauging networks

P2L31: I believe that there is a more current NWWG that could be referred to here (1997).

P4L32: Do any of the numerous Scotty papers document/support this chemically dilute water additions with other indicators (EC, major ions etc)?

Paragraph stating on P6L31: Indeed, the authors have contributed some important findings on peat properties. However, this paragraph seems to be much more about understanding peat physical and hydrological properties than anything specific about Scotty Creek. Although this is fine, the literature discussed is limited to only the work that the authors have done. As the significance of the findings are related to peat properties in general, and the work was largely completed in the lab, it seems prudent to expand beyond exclusively self-citations and include other papers and key findings related to peat properties in general, since this section is not necessarily about Scotty specifically?

P8L34 – P9L7: It is unclear to me the relevance of including this lab-based study on mulching?

P9L9-18: Indeed, here the authors mention the applicability of the findings from Scotty to a large number of other landscapes. This is, in my opinion, a missed opportunity to provide a more comprehensive synthesis of exactly how the decades of research at Scotty are explicitly applicable elsewhere. Expanding on this section could be a valuable opportunity to integrate the Scotty findings into other landscapes more definitively. A well-crafted paragraph here would strengthen this paper and increase the usefulness and applicability beyond Scotty.

P11L24: There is not a fig.8a in this paper, so remove this? If it was referring to a specific Fig in the Quinton paper, it seems odd and is a bit misleading to this reader.

P12L20-25: Has water fluxes through the talik subsurface flowpaths been quantified? How much (mm/y) is this water loss estimated to be? As the authors demonstrate, the saturated hydraulic conductivity of peat decreases rapidly with depth, so perhaps the K of talik peat is low, and limits these water fluxes? More information / quantifying this would be useful here.

Summary/Conclusions section: I found the last few sections of the paper to be a bit laborious to read through. I think that a guiding conceptual figure would be useful to help keep the reader focussed and engaged, and to help someone not as familiar with the wealth of research from Scotty to more easily grasp the big picture and key findings. I do not know what this would look like, but some type of image that provides an overview / demonstrates the key changes to the landscape and their impact on the hydrological function of this region over the 3 decades of study. I think this would be very useful for potential readers.

---

## Referee Comment (RC4) · Anonymous Referee #4 · 8 Oct 2018

This manuscript presents a summary of the research results from two decades of research at the Scotty Creek basin with a focus on the mechanistic understanding of hydrology in discontinuous permafrost systems that has been developed over this time. It provides a valuable resource that pulls together this important body of literature and highlights the importance of long-term, field-based hydrological studies.

Overall, the paper is well-written and easy to follow. There is a huge amount of information presented, but it flows fairly well throughout. I do recommend that the authors take another careful read through as I found some places where grammar could be improved and text could be clarified, but I did not mark them all here. Although somewhat of a personal style choice, several portions of the paper would be clearer if switched to the active voice. It would also be useful to have a map illustrating the location of instrumentation listed in Table 1, possibly as an additional panel(s) to Figure 1. My other main suggestion would be a careful consideration of the use of terminology. In the early sections, the authors refer to collapse scars and later I think these are referred to as flat bogs. Maybe these are not the same thing, but in any case, I wasn't sure, so a clear and consistent use of terminology will help this. A few additional comments are below.

Page 3, line 15: Are you able to provide an estimate of how much of the subarctic this type of land cover accounts for?

Page 5, lines 15-16: And what is the main difference between these subarctic peatlands and those further south? Further south peatland can act as water sources, transmitters or sinks. Can you be more specific here?

Page 8, line 11: I suggest a hyphen between regions and adapted to make the term clearer – so "cold regions-adapted"

Page 11, lines 4-9: This paragraph doesn't report on the findings, but what was done. Either remove it, or edit to tell the reader the main finding from each study.

Page 12, line 2: When the authors refer to bogs in this sentence, is this the same feature described as a collapse scar in the early sections? If so, please use the same terminology here. See also others instances of the term bogs throughout this section.

Page 14, line 5: Here the authors define "bog-capture", but it has already been used on several occasions in the preceding paragraphs. Consider moving this definition forward to the first instance of the term bog-capture.

Page 16, line 20: Do you know the density of the linear disturbance in the catchment? Or the total length? That would be a useful addition here if known.

Page 17, summary and conclusions: This section does a good job of illustrating the importance of the research at Scotty Creek to the broader hydrological community, but could also be used to set the stage for what is to come. What are the most important

remaining knowledge gaps? What should the next 5 years of research priorities look like? Many of the sections of the paper introduce needs for future research, but this section can really highlight what still needs to be done.

---

## Author Comment (AC2) · 23 Nov 2018

Page 2 Line 18: The reference to Figure 2 here is odd, since the figure does not indicate anything about expanding precipitation and stream gauging networks

ć Agreed. Reference to Figure 2 has been removed.

P2L31: I believe that there is a more current NWWG that could be referred to here (1997).

ć Agreed. The reference has been updated to "(NWWG, 1997)".

P4L32: Do any of the numerous Scotty papers document/support this chemically dilute water additions with other indicators (EC, major ions etc.)?

ć Electrical conductance was found to increase with increasing distance from the edge of plateaus. This EC data was not published, but this observation will be noted in the paper with reference to “unpublished data”.

Paragraph starting on P6L31: Indeed, the authors have contributed some important findings on peat properties. However, this paragraph seems to be much more about understanding peat physical and hydrological properties than anything specific about Scotty Creek. Although this is fine, the literature discussed is limited to only the work that the authors have done. As the significance of the findings are related to peat properties in general, and the work was largely completed in the lab, it seems prudent to expand beyond exclusively self-citations and include other papers and key findings related to peat properties in general, since this section is not necessarily about Scotty specifically?

ć Agreed. This paragraph will be expanded to include other papers and key findings related to peat physical and hydraulic properties.

P8L34 – P9L7: It is unclear to me the relevance of including this lab-based study on mulching?

ć The sentence starting on line 34 (“Further climate chamber. . .”) and the remaining sentences of that paragraph have been separated from the text above. This new paragraph will focus on the development and testing of thaw mitigation strategies at Scotty Creek. This will include new text on novel new thermosiphon designs and applications, mulching, snow detectors, and other techniques and devices developed and tested at Scotty Creek.

P9L9-18: Indeed, here the authors mention the applicability of the findings from Scotty to a large number of other landscapes. This is, in my opinion, a missed opportunity to provide a more comprehensive synthesis of exactly how the decades of research at Scotty are explicitly applicable elsewhere. Expanding on this section could be a valuable opportunity to integrate the Scotty findings into other landscapes more definitively.

A well-crafted paragraph here would strengthen this paper and increase the usefulness and applicability beyond Scotty.

ć This section will be moved to the Summary and Conclusions section where it will be expanded upon.

P11L24: There is not a fig.8a in this paper, so remove this? If it was referring to a specific Fig in the Quinton paper, it seems odd and is a bit misleading to this reader.

ć This reference was intended to guide the reader not just to a specific paper, but to a specific figure in that paper. We have re-expressed this reference to read: "(Fig. 8a in Quinton et al., 2009a)". However, if the reviewer and/or the Editor does not like this reference style, we are happy to just refer the paper only and not also to a specific part of it.

P12L20-25: Has water fluxes through the talik subsurface flowpaths been quantified? How much (mm/y) is this water loss estimated to be? As the authors demonstrate, the saturated hydraulic conductivity of peat decreases rapidly with depth, so perhaps the K of talik peat is low, and limits these water fluxes? More information / quantifying this would be useful here.

ć This is the subject of a current PhD student (E. Devoie). Some preliminary results are available based on the use of passive flux metres during the summer of 2018 that followed the initial submission of this manuscript. We plan to describe some of these initial findings in the revised draft of the present manuscript.

Summary/Conclusions section: I found the last few sections of the paper to be a bit laborious to read through. I think that a guiding conceptual figure would be useful to help keep the reader focussed and engaged, and to help someone not as familiar with the wealth of research from Scotty to more easily grasp the big picture and key findings. I do not know what this would look like, but some type of image that provides an overview / demonstrates the key changes to the landscape and their impact on the

hydrological function of this region over the 3 decades of study. I think this would be very useful for potential readers.

ć We are preparing a conceptual model that depicts 1) the accumulated understanding of the hydrological functioning of the major land-cover types that predominate the Scotty Creek basin and surrounding region, and 2) the trajectory of the permafrost thaw-induced land cover and hydrological change.

---

## Author Comment (AC3) · 23 Nov 2018

It would also be useful to have a map illustrating the location of instrumentation listed in Table 1, possibly as an additional panel(s) to Figure 1.

ć A map showing the location of the infrastructure referred to in the Table has been prepared. It will be included as Figure 1c.

My other main suggestion would be a careful consideration of the use of terminology. In the early sections, the authors refer to collapse scars and later I think these are referred to as flat bogs. Maybe these are not the same thing, but in any case, I wasn't sure, so a clear and consistent use of terminology will help this. A few additional comments are below.

ć We thank the reviewer for pointing this out. “Flat bog” is a term we used in earlier publications. However, its use was problematic because, strictly speaking, the peat plateaus are also bogs since they receive water from precipitation only, and as such are similar to the well-described “domed bogs” found at lower latitudes. We have revised the present manuscript to remove the term “flat bogs” (except to explain that it was used in earlier papers) and have replaced it with “collapse scar bogs” or simply “collapse scars”.

Page 3, line 15: Are you able to provide an estimate of how much of the subarctic this type of land cover accounts for?

ć An estimate of the extent of Scotty-like terrain is possible in the southern Taiga Plains, such as in the area captured in Fig. 1a. We will include such an estimate in the revised draft.

Page 5, lines 15-16: And what is the main difference between these subarctic peatlands and those further south? Further south peatland can act as water sources, transmitters or sinks. Can you be more specific here?

ć The main difference is that the peat plateaus are relatively impermeable to infiltrating water owing to the presence of saturated permafrost close to the ground surface. Raised bogs at lower latitudes are without permafrost and are therefore more permeable. As such they have a greater capacity to store water and therefore shed less runoff than peat plateuas. The presence of permafrost below the peat plateaus also decouples the overlying active layer from interacting with groundwater. Also, the juxtaposition of permafrost and permafrost-free terrains in the study region results in greater contrast in the hydrological functions of adjacent terrain types (e.g. peat plateau and adjacent collapse scar) at the study site compared to peatland terrains at lower latitudes. This explanation will be added to lines 15-16 of page 5 in the revised manuscript.

Page 8, line 11: I suggest a hyphen between regions and adapted to make the term clearer – so “cold regions-adapted”

ć Agreed. Done.

Page 11, lines 4-9: This paragraph doesn't report on the findings, but what was done. Either remove it, or edit to tell the reader the main finding from each study.

ć The purpose of this paragraph was to show that new knowledge on water flux and storage processes was not an end in itself, but has been used to inform hydrological model development so that hydrological predictions can be made more confidently. For example, such models are important tools for evaluating hydrological impacts for different scenarios of permafrost thaw-induced land-cover change. We will also add text explaining how the CRHM model (informed by process studies at Scotty Creek) runs in the Stone et al. shed light on the nature of hydrological interactions among the major land cover types.

Page 12, line 2: When the authors refer to bogs in this sentence, is this the same feature described as a collapse scar in the early sections? If so, please use the same terminology here. See also others instances of the term bogs throughout this section.

ć Yes, this is the same feature. The text has been revised so that "flat bog" and "bog' are replaced by "collapse scar bog" or simply "collapse scar".

Page 14, line 5: Here the authors define "bog-capture", but it has already been used on several occasions in the preceding paragraphs. Consider moving this definition forward to the first instance of the term bog-capture.

ć The definition has been moved so that it accompanies the first instance of its use in the paper.

Page 16, line 20: Do you know the density of the linear disturbance in the catchment? Or the total length? That would be a useful addition here if known.

ć The density of linear disturbance at Scotty Creek is approximately 1 km of disturbance per 1 km2. This is approximately 7 times the density of the drainage network of channel fens and open channels. This information will be added to the revised

manuscript.

Page 17, summary and conclusions: This section does a good job of illustrating the importance of the research at Scotty Creek to the broader hydrological community, but could also be used to set the stage for what is to come. What are the most important remaining knowledge gaps? What should the next 5 years of research priorities look like? Many of the sections of the paper introduce needs for future research, but this section can really highlight what still needs to be done.

ć This point was also raised by the other reviewers. We are preparing a conceptual model that depicts 1) the accumulated understanding of the hydrological functioning of the major land-cover types that predominate the Scotty Creek basin and surrounding region, and 2) the trajectory of the permafrost thaw-induced land cover and hydrological change. This will help to set the stage for what is to come – i.e. present research gaps and research needs for the future.

---

## Author Response (AR1)

**Report on revisions to "A Synthesis of Three Decades of Eco-Hydrological Research at Scotty Creek, NWT, Canada"**

Each comment provided by the reviewers is listed below. Below each comment, the response of the authors is provided. Page (P) and line numbers refer to those of the original manuscript. This is followed by a version of the manuscript which identifies each change that was made between the initial submitted draft and the revised draft.

**Reviewer 1.**

P2 Line 18 discusses precipitation and stream gauging networks with reference to figure 2. The expectation is of a map that shows gauging sites, but this is missing.

- The types of data collected at Scotty Creek are listed in Table 1. The Table 1 caption now reads: Table 1: "Instrumentation at Scotty Creek Research Site. A map showing the locations of scientific and other infrastructure at the Scotty Creek Research Station is available at http://www.scottycreek.com/. This on-line map is regularly updated".

P3 section 2. This is a paper about hydrological functioning, but no basic data are provided to introduce the climate context for the basin. We need a brief summary of the climatic setting and precipitation characteristics, and a sentence or so on the flow regime.

- Precipitation has been measured at Fort Simpson, 50 km north of Scotty Creek since 1896. However, precipitation and streamflow data records at Scotty began only in 1996. The reviewer makes a good point, so we added streamflow data from the adjacent Jean Marie River (1972-present) to characterise in a few sentences, the flow regimes for the study region.

P3 line 2 delete 'through this process'. It's not clear what this process is and the phrase is in any case unnecessary.

- 'Through this process' has been removed.

P4 line 9 delete comma

- deleted.

P4 Line 13 maybe 'primary hydraulic function' - there are many functional aspects of these systems

- 'primary function' changed to 'primary hydraulic function'.

P5 line 2 insert comma after 2003)

- Comma added

P5 Lines 14, 16, 17 – too many ‘unique’s – poor English.

- These sentences have been revised as follows:

  “Collectively, these studies helped to define the hydrological functions of the land cover types that predominate not only at Scotty Creek, but throughout much of the peatland-dominated zone of discontinuous permafrost. These studies also illustrate the unique hydrological behaviour of sub-arctic peatlands compared with their counterparts in warmer climates.”

P6 line 4 ‘total porosity decreases by 10%’ - please be specific, is this a 10% decrease in the porosity value, or is the porosity expressed as a percentage changing by 10? There is a use of undefined terminology here – what is meant by water storage coefficient, and active porosity? How is active porosity different from drainable porosity?

- To address these comments, the paragraph was re-written as follows:

  “Quinton *et al*. (2000) reported that the values of total porosity decreased only slightly with depth, from approximately 95% in the 0-5 cm zone to 85% at 35 cm depth. However, the authors reported that over this depth range, values of ‘active porosity’ (*i.e.* the proportion of the total peat pore volume that actively transmits water (Romanov, 1968)) typically decreases from approximately 80% to <50%. The authors measured active porosity from image analysis of 2D thin sections of peat samples, whereby the inter-particle area expressed as a percentage of the total image area was assumed to approximate the active porosity. Quinton and Hayashi (2004) also demonstrated that the fraction of the total porosity that conducts water decreases with increasing depth, although these authors did so through drainage experiments. They reported that the ‘drainable porosity’ (*i.e.* the pore volume of water removed when the water table is lowered) decreases from approximately 0.6 near the ground surface to 0.05 at 40 cm depth. These relatively low values, especially at depth, enable a rapid response of the water table to hydraulic inputs to the ground surface. This rapid response is enhanced by very high infiltration rates and the close proximity of the saturated zone to the ground surface. Laboratory drainage tests combined with microscopic image analyses indicate that with increasing depth, the proportion of small, closed and dead-end pores increases as does the water content for a given pressure, and that the peat maintains a residual volumetric moisture content of 15-20% (Quinton and Hayashi, 2004).”

How is a decrease of ‘only 10%’ consistent with a ‘rapidly decreasing’ storage coefficient?

- The different expressions of porosity (total, active, inactive) were used to demonstrate that the ratio of water that can flow and that which remains in storage changes with depth such that the storage fraction increases with depth. That is what was meant by the depth variation in the storage coefficient. However, we agree with the reviewer and have taken his/her advice and provided more explanation in the paragraph (above). The reason for the greater storage of water with increasing depth is due to the decreasing ability with depth of the peat to convey water, as measured from image analysis (active porosity) and drainage tests (drainable porosity).

P6 Line 18 k is loosely defined here as permeability. Presumable intrinsic permeability is what is meant.

- Correct. We are referring to the soil permeability [$L^2$] which depends only on the properties of the medium, *e.g.* the size, shape, number, distribution and continuity of the conducting pores. Freeze and Cherry (1979) suggest the approximation, $k = cD^2$; where c, is a dimensionless coefficient and D, is the pore or grain diameter.

P9 line 25 retained rather than detained

- The change has been made.

P10 line 5 delete comma after 2003, insert comma after 'that'

- done

p10 line 30 the SFASH model p12 line 20 insert comma after plateaus,

- done

P16 line 7 what precisely is meant by a composite hydrograph? Is this the mean over the period, or something different?

- Agreed. The word "composite" was removed and replaced with the word "mean".

P16 line 28 what precisely is meant by 'soil moisture' here? It's difficult to see how total moisture would increase if porosity decreases.

- This sentence has been modified as follows:

  "The initial disturbance displaces the ground surface downward where it is closer to the water table. As a result, the soil moisture content (and therefore the bulk thermal conductivity) of the peat near the ground surface increases, and preferential ground thaw is initiated."

P17. Summary. Please summarize conclusions for basin storage. Only flows are discussed.

- The summary has been revised so that it includes conclusions for both flow and storage processes.

P18 the paper ends quite abruptly. We need a section of reflection here. After such extensive filed work over a long period, what are the outstanding gaps in knowledge and primary requirements for future work?

- The conclusion was revised so that it includes discussion of outstanding gaps in knowledge and primary requirements for future work.

**Reviewer 2.**

My main criticism is that this article could be alternatively titled 'My life's work'. There are 40 self-citations to the first author's work alone, which is highly unusual. However, I can appreciate that there is no real way to circumvent this, as doubtless the authors of this paper (especially the first author) have led the work at this site for many years and produced many important papers. However, as a comment, I wonder if they could include a section (or at least some more text) relating the work at Scotty Creek to other long-term northern, discontinuous permafrost research catchments, even ones that may have different physical conditions. That might give this broader appeal.

- We thank the reviewer for these suggestions. However we feel that a title such as "My life's work" for a research programme that has involved such a large number of research colleagues, local community members, and hundreds of students, would be grossly inaccurate. We do agree that it is unusual to see a large number of "self-citations", however, it is also unusual for researcher to dedicate his entire career to a building a research station for long term (*i.e.* multi-decadal) studies. It is the study site that is the subject of this paper. We have follow the referee's suggestion of adding text relating the work at Scotty to other long-term research observatories, including a new section on the impact of Scotty Creek research on the larger research community.

2. I don't find the first two paragraphs of the introduction particularly well written. In my opinion the first paragraph should talk about circum-polar subarctic cold regions/ permafrost/hydrology work to begin with (at a large-scale) and then narrow to the region containing Scotty Creek. This paper starts with a rather abrupt introduction to the Liard River valley, and then expands in the second paragraph to the broader Mackenzie River basin. Restructuring and enhancing the circum-polar questions and implications would position this paper better in the broader literature. I think that is important, as it is certainly not standard to write a review paper in a hydrology journal that is focused on just one site, and it would be useful for the authors to really emphasize the critical nature of this work.

- We thank the reviewer for these comments. We rewrote the first two paragraphs of the paper so that it starts off with a broader scope, which we think has strengthened the paper.

3. Sections 2 and 3 are a bit of a laborious read and describe processes and concepts that could easily be visually represented. Why not have a figure in Section 2 that shows the landscape units and how they function and interact hydrologically? This would be useful when the authors refer to the 'new conceptual framework for runoff generation' (P5, L23). Some of this would be easier to see than read (although I recommend that the text be retained). Section 3 could have a figure that shows some active layer conditions or processes (e.g. data for K vs. depth or some freeze-thaw conditions).

- A conceptual diagram was added to the figures (Figure 12) and is the focus of the final section of the manuscript ("Ongoing and future studies") which synthesises the research of the previous section.

4. I miss much discussion on biogeochemical processes or storage and lateral fluxes of nutrients etc. I understand that has not been the focus at Scotty Creek, but I am surprised there are so few studies to refer to. Related to this, there is very little discus- sion on vegetation changes at Scotty Creek except where the focus is the impacts on hydrologic processes. Surely some of the Baltzer work could be better highlighted. I would suggest giving more than a passing glance to the important Baltzer et al. 2014 GCB (P11, L26) study. Although Eco-Hydrological is included in the title, there surely isn't much eco content in the paper.

- It was decided at the outset to limit the scope of this paper to the 20+ years of hydrological research. The lad author is aware of the others studies at Scotty Creek (*e.g.* studies that focus on GHG, ecology, sedimentology, biogeochemistry, etc.), since he invited the researchers to pursue such studies. However, the scope of the paper would be far too broad, and the length too long. This would be better accomplished with a Scotty Creek book, with separate chapters detailing the advances of different areas of study. This has been done for other long term stations (*e.g.* Wolf Creek, Yukon).

5. Figure 3 on the active layer and talik etc. could also include an additional pane or two showing the 'dual layered system' of flow (P12, L14).

- This has been done.

6. How much longer will the increased hydrologic connectivity result in higher flows? If the wetlands are dewatering, when will the basin reach 'peak water'?? It seems to me that this concept related to glacier discharge could also be applied to Scotty Creek as the wetland storage decreases.

- Since this paper was submitted, we have advanced our understanding of the permanence of water storage changes in response to permafrost thaw. This advancement is discussed in the last two sections of the revised draft.

7. The conclusion ends abruptly. It almost seems like text were deleted by accident. There is no vision, no path forward, no high-level statement explaining the importance of the work. Something needs to be rewritten here.

- This was also noted by other reviewers. This conclusion dection was divided into two sections, one focussed on the impacts of research at Scotty Creek on the larger scientific community, and the other on on-going and future research at Scotty Creek. For the latter, a new conceptual diagramme (Figure 12) was added to the paper as requested.

P2, L5, should ‘regions’ be after ‘warming’?

- This sentence has been reworded.

Figure 1, this is obvious to Canadians, but probably ‘Canada’ could be labeled in the inset

- Done.

**Reviewer 3.**

Page 2 Line 18: The reference to Figure 2 here is odd, since the figure does not indicate anything about expanding precipitation and stream gauging networks

- Reference to Figure 2 has been removed.

P2L31: I believe that there is a more current NWWG that could be referred to here (1997).

- We used the 1988 edition because it contains the detailed descriptions referred to in this paper. Although the 1997 edition is more recent, referring to it would be a less direct path for the reader to follow to arrive at the points referred to in this paper.

P4L32: Do any of the numerous Scotty papers document/support this chemically dilute water additions with other indicators (EC, major ions etc.)?

- Electrical conductance was found to increase with increasing distance from the edge of plateaus. This EC data was not published, but this observation was added to the paper with reference to “unpublished data”.

Paragraph starting on P6L31: Indeed, the authors have contributed some important findings on peat properties. However, this paragraph seems to be much more about understanding peat physical and hydrological properties than anything specific about Scotty Creek. Although this is fine, the literature discussed is limited to only the work that the authors have done. As the significance of the findings are

related to peat properties in general, and the work was largely completed in the lab, it seems prudent to expand beyond exclusively self-citations and include other papers and key findings related to peat properties in general, since this section is not necessarily about Scotty specifically?

- This paragraph was expanded to include other papers and key findings related to peat physical and hydraulic properties.

P8L34 – P9L7: It is unclear to me the relevance of including this lab-based study on mulching?

- The sentence starting on line 34 (“Further climate chamber…”) and the remaining sentences of that paragraph have been separated from the text above. This new paragraph focusses on the development and testing of thaw mitigation strategies at Scotty Creek. This includse new text on novel ground freezing systems including new thermosiphon designs and applications, mulching, snow detectors, and other techniques and devices developed and tested at Scotty Creek.

P9L9-18: Indeed, here the authors mention the applicability of the findings from Scotty to a large number of other landscapes. This is, in my opinion, a missed opportunity to provide a more comprehensive synthesis of exactly how the decades of research at Scotty are explicitly applicable elsewhere. Expanding on this section could be a valuable opportunity to integrate the Scotty findings into other landscapes more definitively. A well-crafted paragraph here would strengthen this paper and increase the usefulness and applicability beyond Scotty.

- This was moved and incorporated into the final two section of the revised paper where it was expanded upon.

P11L24: There is not a fig.8a in this paper, so remove this? If it was referring to a specific Fig in the Quinton paper, it seems odd and is a bit misleading to this reader.

- This reference was intended to guide the reader not just to a specific paper, but to a specific figure in that paper. We have re-expressed this reference to read: “(Fig. 8a in Quinton et al., 2009a)”. Without this explicit reference, it would be more time consuming for the reader to locate the specific material referred to here.

P12L20-25: Has water fluxes through the talik subsurface flowpaths been quantified? How much (mm/y) is this water loss estimated to be? As the authors demonstrate, the saturated hydraulic conductivity of peat decreases rapidly with depth, so perhaps the K of talik peat is low, and limits these water fluxes? More information / quantifying this would be useful here.

- This is the subject of a current PhD student (E. Devoie). Some preliminary results are available based on the use of passive flux metres during the summer of 2018 that followed the initial submission of this manuscript. We have described some of these key initial findings in the revised draft.

Summary/Conclusions section: I found the last few sections of the paper to be a bit laborious to read through. I think that a guiding conceptual figure would be useful to help keep the reader focussed and engaged, and to help someone not as familiar with the wealth of research from Scotty to more easily grasp the big picture and key findings. I do not know what this would look like, but some type of image that provides an overview / demonstrates the key changes to the landscape and their impact on the hydrological function of this region over the 3 decades of study. I think this would be very useful for potential readers.

- We have added a conceptual model that depicts 1) the accumulated understanding of the hydrological functioning of the major land-cover types that predominate the Scotty Creek basin and surrounding region, and 2) the trajectory of the permafrost thaw-induced land cover and hydrological change.

**Reviewer 4.**

It would also be useful to have a map illustrating the location of instrumentation listed in Table 1, possibly as an additional panel(s) to Figure 1.

- The types of data collected at Scotty Creek are listed in Table 1. The Table 1 caption now reads: Table 1: "Instrumentation at Scotty Creek Research Site. A map showing the locations of scientific and other infrastructure at the Scotty Creek Research Station is available at http://www.scottycreek.com/. This on-line map is regularly updated".

My other main suggestion would be a careful consideration of the use of terminology. In the early sections, the authors refer to collapse scars and later I think these are referred to as flat bogs. Maybe these are not the same thing, but in any case, I wasn't sure, so a clear and consistent use of terminology will help this. A few additional comments are below.

- We thank the reviewer for pointing this out. "Flat bog" is a term we used in earlier publications. However, its use was problematic because, strictly speaking, the peat plateaus are also bogs since they receive water from precipitation only, and as such are similar to the well-described "domed bogs" found at lower latitudes. We have revised the present manuscript to remove the term "flat bogs" (except to explain that it was used in earlier papers) and have replaced it with "collapsed wetlands".

Page 3, line 15: Are you able to provide an estimate of how much of the subarctic this type of land cover accounts for?

- An estimate of the extent of Scotty-like terrain is possible in the southern Taiga Plains, such as in the area captured in Fig. 1a. We have include estimated values in the revised draft.

Page 5, lines 15-16: And what is the main difference between these subarctic peatlands and those further south? Further south peatland can act as water sources, transmitters or sinks. Can you be more specific here?

- The main difference is that the peat plateaus are relatively impermeable to infiltrating water owing to the presence of saturated permafrost close to the ground surface. Raised bogs at lower latitudes are without permafrost and are therefore more permeable. As such they have a greater capacity to store water and therefore shed less runoff than peat plateuas. The presence of permafrost below the peat plateaus also decouples the overlying active layer from interacting with groundwater. Also, the juxtaposition of permafrost and permafrost-free terrains in the study region results in greater contrast in the hydrological functions of adjacent terrain types (*e.g.* peat plateau and adjacent collapse scar) at the study site compared to peatland terrains at lower latitudes. This explanation was added to lines 15-16 of page 5.

Page 8, line 11: I suggest a hyphen between regions and adapted to make the term clearer – so "cold regions-adapted"

- Agreed. Done.

Page 11, lines 4-9: This paragraph doesn't report on the findings, but what was done. Either remove it, or edit to tell the reader the main finding from each study.

- The purpose of this paragraph was to show that new knowledge on water flux and storage processes was not an end in itself, but has been used to inform hydrological model development so that hydrological predictions can be made more confidently. For example, such models are important tools for evaluating hydrological impacts for different scenarios of permafrost thaw-induced land-cover change. We added text explaining how the CRHM model (informed by process studies at Scotty Creek) runs in the Stone *et al.* shed light on the nature of hydrological interactions among the major land cover types.

Page 12, line 2: When the authors refer to bogs in this sentence, is this the same feature described as a collapse scar in the early sections? If so, please use the same terminology here. See also others instances of the term bogs throughout this section.

- Yes, this is the same feature. The text has been revised so that "flat bog" and "bog' are replaced by "collapsed wetland".

Page 14, line 5: Here the authors define "bog-capture", but it has already been used on several occasions in the preceding paragraphs. Consider moving this definition forward to the first instance of the term bog-capture.

- The definition has been moved so that it accompanies the first instance of its use in the paper.

Page 16, line 20: Do you know the density of the linear disturbance in the catchment? Or the total length? That would be a useful addition here if known.

- The density of linear disturbance at Scotty Creek is approximately 1 km of disturbance per 1 $km^2$. This is approximately 7 times the density of the drainage network of channel fens and open channels. This information was added to the revised manuscript.

Page 17, summary and conclusions: This section does a good job of illustrating the importance of the research at Scotty Creek to the broader hydrological community, but could also be used to set the stage for what is to come. What are the most important remaining knowledge gaps? What should the next 5 years of research priorities look like? Many of the sections of the paper introduce needs for future research, but this section can really highlight what still needs to be done.

- This point was also raised by the other reviewers. We are prepared a conceptual model that depicts 1) the accumulated understanding of the hydrological functioning of the major land-cover types that predominate the Scotty Creek basin and surrounding region, and 2) the trajectory of the permafrost thaw-induced land cover and hydrological change. This new figure (Figure 12) is the focus of discussion in the final section (On-going and future studies) of the revised paper.

[revised manuscript text omitted]

---

## Author Response (AR2)

Editor Decision: Publish subject to minor revisions (review by editor) (19 Mar 2019) by Chris DeBeer

Comments to the Author:
Dear authors,

Thank you for your efforts to respond to and address the reviewer comments. The paper has been improved as a result and is nearly ready for final publication. I have a few minor comments that can be addressed fairly easily.

P13, L13: the word "hydrological" is written twice.

- This was fixed.

P17, L10: Why is Fig. 2 referred to here? Should this be Fig. 9?

- Changed to Figure 9.

P16-17 and Fig. 9: It would be helpful to explain why the adjusted precipitation data (https://www.canada.ca/en/environment-climate-change/services/climate-change/science-research-data/climate-trends-variability/adjusted-homogenized-canadian-data/precipitation.html) for Ft. Simpson were not used. These data do show an increasing trend in total annual precipitation. How can we be sure that the observed increase in basin discharge is not at least partly due to an increase in precipitation?

- An explanation was provided and this was backed-up by a new figure (Figure 9b).

Fig. 8: What is pre-1996 R.R. and what is the shaded area on the graph meant to represent? This is described in Haynes et al. (2018), but should be explained here as well for clarity.

- Both pre-1996 R.R. and shaded area are now defined and explained in an expanded figure caption.

P20, L3: Are the units correct? (km km^-1 – if this is total length divided by basin area, should it be km km^-2?)

- Units changed to km km^-2

Section 8: My only other comment is that while it is helpful to have added Section 8 on “ongoing and future studies”, this section mostly reads as though it is a continuation of the process descriptions on hydrological functioning and future system trajectory, mainly belonging in sections 3, 4, and 5. There is a need here, as the reviewers were calling for, to include a summary of gaps, outstanding research questions and needs, and primary requirements for future work. For instance, presently there is only this single sentence: “The rates and patterns of this regeneration, the environmental processes and feedbacks affecting it, and its hydrological implications are areas requiring further study.”

- Section 8 was expanded to include a summary of gaps, outstanding research questions and needs, and primary requirements for future work for each subject discussed.

[revised manuscript text omitted]